# Endemism patterns are scale dependent

Barnabas H. Daru [1✉], Harith Farooq[2,3,4,5], Alexandre Antonelli [2,3,6] & Søren Faurby [2,3]

Areas of endemism are important in biogeography because they capture facets of biodiversity not represented elsewhere. However, the scales at which they are relevant to research and conservation are poorly analysed. Here, we calculate weighted endemism (WE) and phylogenetic endemism (PE) separately for all birds and amphibians across the globe. We show that scale dependence is widespread for both indices and manifests across grain sizes, spatial extents and taxonomic treatments. Variations in taxonomic opinions—whether species are treated by systematic 'lumping' or 'splitting'—can profoundly affect the allocation of WE hotspots. Global patterns of PE can provide insights into complex evolutionary processes but this congruence is lost at the continental to country extents. These findings are explained by environmental heterogeneity at coarser grains, and to a far lesser extent at finer resolutions. Regardless of scale, we find widespread deficits of protection for endemism hotspots. Our study presents a framework for assessing areas for conservation that are robust to assumptions on taxonomy, spatial grain and extent.

[1] Department of Life Sciences, Texas A&M University-Corpus Christi, Corpus Christi 78412 TX, USA. [2] Department of Biological and Environmental Sciences, University of Gothenburg, SE 405 30 Gothenburg, Sweden. [3] Gothenburg Global Biodiversity Centre, Box 461, SE 40530 Gothenburg, Sweden. [4] Department of Biology & CESAM, University of Aveiro, Aveiro, Portugal. [5] Faculty of Natural Sciences at Lúrio University, Cabo Delgado, Mozambique. [6] Royal Botanic Gardens, Kew, TW9 3AE, Richmond, Surrey, UK. ✉email: barnabas.daru@tamucc.edu

Biodiversity patterns and their underlying mechanistic processes are inherently scale dependent[1–4]. Patterns and processes predicted at one spatial scale may not necessarily be predicted at other scales. Information such as landscape heterogeneity can be lost at coarser spatial scales[5,6], while properties such as those caused by speciation dynamics may emerge[7]. Several studies have indicated that scale dependence may be pervasive in patterns of species richness[3,8–13], density dependence[14–16], extinction risk[17], ratios of native/exotic species[18] or migration and colonization rates[19,20]. It has also been suggested that the effects of scale may be common in patterns of endemism[21–26], yet there has never been, to the best of our knowledge, a global assessment of this phenomenon.

Two important spatial metrics of endemism are weighted endemism and phylogenetic endemism. Weighted endemism is species richness inversely weighted by species ranges[27,28]. Phylogenetic endemism is the phylogenetic equivalent of species endemism and is measured as the total phylogenetic branch length spanned by species in an area, after dividing each branch length by the global range size of its descendant clade (measured in Myr/km$^2$)[29]. This means that even areas with widespread species are detected as areas of phylogenetic endemism if the local communities are phylogenetically overdispersed. Just as the two metrics capture different facets of endemism and are increasingly considered crucial for conservation prioritization[30–32], the effect of scale is also expected to vary differently among them. This is because both metrics depend on spatial grain (i.e. resolution), extent[10] and/or taxonomic treatment[33]. Weighted endemism can be sensitive to changes in taxonomic opinion because small-ranged species are weighted equally. Advances in taxonomic knowledge lead to continuous changes in the number and delineation of species, either through lumping several species into one or splitting single species into several[34,35]. For instance, over the past 110 years, bird species have witnessed varying estimates in their numbers: 18,939 species in 1909[36], 8590 in 1951[37] and 10,738 species today[38]. Such changes in taxonomic concepts can influence estimates of weighted endemism, and by consequence bias, undermine or obscure any underlying evolutionary mechanisms[39,40].

In contrast, phylogenetic endemism offers a potential solution to deal with new taxonomic knowledge in conservation strategies. This is because phylogenetic endemism is not greatly influenced by oversplitting of neoendemics (more phylogenetically derived species)—for example, if populations only isolated since the last ice age are elevated to species level. Patterns of phylogenetic endemism tend to manifest at large global extents, but phylogenetic endemism can be severely influenced in a continental or country setting[31,41]. For example, the Galápagos penguin (*Spheniscus mendiculus*) is the only penguin occurring naturally outside the Southern Hemisphere, endemic to the Galápagos Islands north of the equator[42]. Assuming all else is equal, its phylogenetic endemism is expected to be higher at a continental extent north of the equator, but lower in a global setting because its closest relatives comprise a group of about 20 species exclusive to the Southern Hemisphere. Incomplete sampling (i.e. missing taxa) or randomly added taxa on the phylogeny (as is often done in macroecological studies) could potentially inflate estimates of phylogenetic endemism[43], in which case taxonomic effects could potentially accumulate. We therefore predict that phylogenetic endemism should vary strongly with spatial extent (Fig. 1), whereas weighted endemism should vary depending on taxonomic conclusions i.e. whether the group has been subject to primarily splitting or lumping (Fig. 1).

Areas of endemism represent important units for postulating hypotheses in historical biogeography[44–46], and are priority targets for conservation action because they capture facets of biodiversity not represented elsewhere[31,32,47,48]. For example, areas that have experienced higher historical temperature change tend to harbour fewer endemic species, often with phylogenetically derived species (neoendemics) occupying higher latitudes[49,50]. In contrast, climatic shifts that lead to low levels of change in species' geographical distributions may allow the survival of ancient lineages that have become extinct elsewhere (paleoendemics)[51]. Therefore, we predict that the local extinction of a paleoendemic lineage can increase patterns of phylogenetic endemism, whereas the loss of a neoendemic will have less impact on phylogenetic endemism, at least initially. Only by losing entire clades will the loss of neoendemics result in a significant change in phylogenetic diversity. A high dispersal rate will cause fewer species to be confined to a specific area, leading to a lower concentration of endemic species[49]. Conversely, the phylogenetic composition of communities including species with poor dispersal abilities will cause the aggregation of close relatives, leading to increased phylogenetic endemism[52].

Despite these considerations, the spatial scales at which areas of endemism are relevant to research and conservation are not well known. Because different groups of organisms differ in their dispersal abilities and home ranges, they are likely to differ greatly in their use of habitat at different spatial grains and extents[53]. Species with wide dispersal capabilities might reflect large geographic range sizes[54], whereas narrow-ranged species may correlate with fine-grained habitat richness[13]. For example, birds have diversified to occupy various habitats and functional roles across most terrestrial and aquatic ecosystems. They show distinct geographic variation in phylogenetic diversity[55,56], and concentrations of spatially restricted phylogenetic diversity have been identified for some clades[32]. On the other hand, amphibians are poor dispersers and possess reduced geographic ranges compared with birds[57], and thus we predict the effect of scale on endemism in amphibians to change across spatial grain and extent.

Here, we use comprehensive datasets on the phylogenetic relationships and geographic distributions for c. 10,000 species of birds and 6000 species of amphibians across the globe to test the hypothesis that changes in taxonomic treatment, spatial grain and extent can influence patterns of weighted endemism and phylogenetic endemism. Specifically, we assess the effects of variations of scale in spatial grain (50, 100, 200, 400 and 800 km), extent (global, continental and national), and taxonomic treatment (based on species' divergence times from 1, 2, 3 to 5 million years ago (Ma)) with respect to the identification of hotspots of weighted and phylogenetic endemism. Our definition of scale encompasses three components: grain, extent and taxonomic treatment. Specifically, we ask three questions: (i) how do patterns of weighted and phylogenetic endemism of different vertebrate clades vary across scales? (ii) at what spatial grain and extent does heterogeneity in environmental factors influence patterns of endemism? and (iii) how effective are the global systems of protected areas in representing hotspots of endemism across grain sizes, spatial extent and taxonomic treatment?

Our results indicate that patterns of weighted and phylogenetic endemism are sensitive to variations in grain sizes, spatial extents and taxonomic treatment, which suggests that their relevance to biogeography and conservation might have been compromised.

## Results and discussion

**Weighted endemism depends on taxonomic treatment.** Using five variations in spatial grain (50, 100, 200, 400 and 800 km), we evaluated changing spatial grain in bird and amphibian weighted endemism based on different extents of taxonomic lumping. To approximate the effect of lumping we successively sliced the phylogenetic tree at various time depths (from 1, 2, 3, to 5 Ma), collapsed nodes and ranges that originated at each time depth,

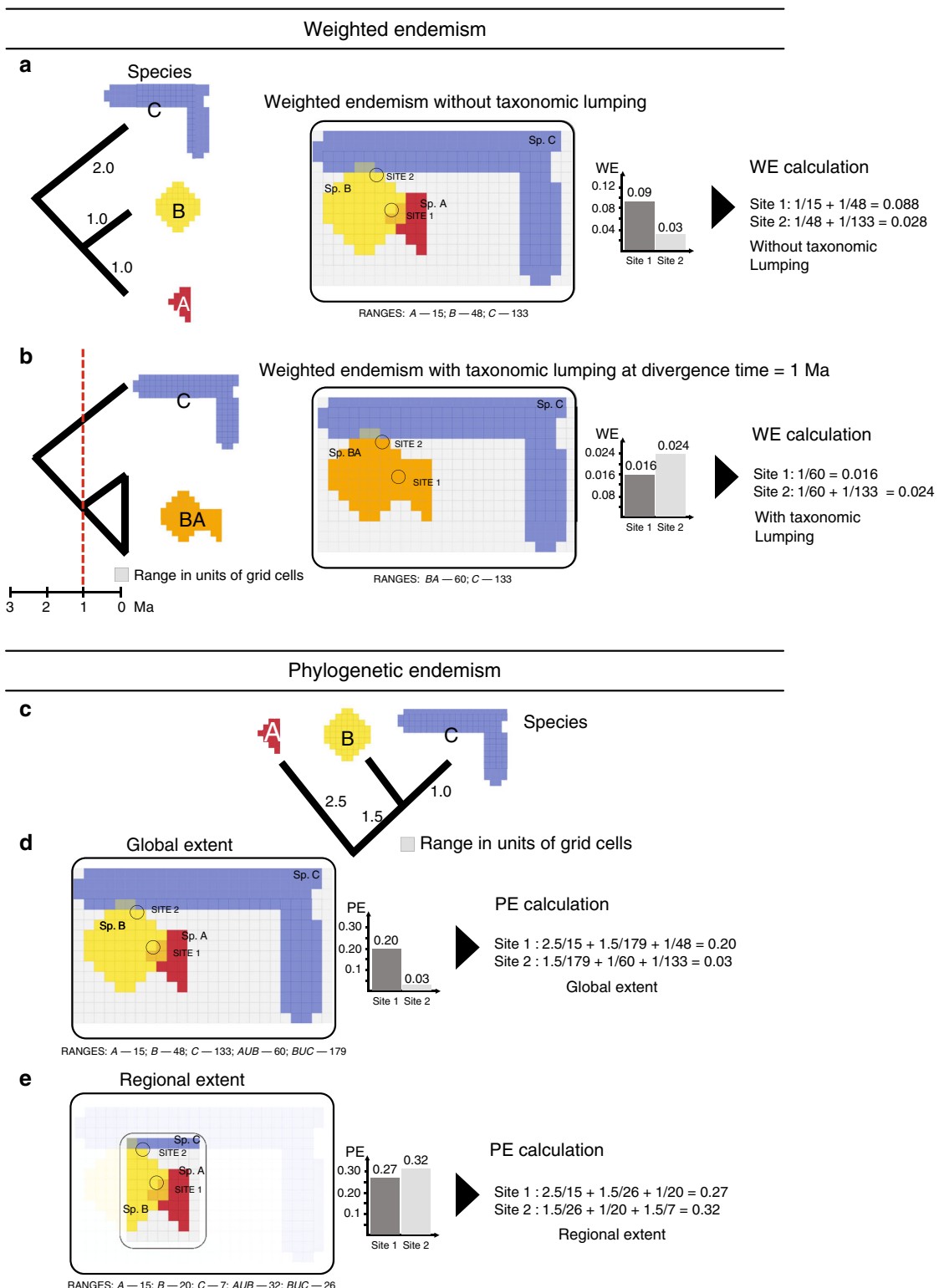

**Fig. 1 Schematic of the variation of endemism with spatial extent, grain size and taxonomic treatment. a**, **b** Changes in weighted endemism (species richness inversely weighted by species ranges) under a scenario of taxonomic lumping at a divergence time of 1 million years ago (Ma). As the node and ranges that originated at 1 Ma were collapsed (**b**), new hotspot maps of weighted endemism were generated which we compared with the original data (**a**). **c** Variation of phylogenetic endemism (the degree to which phylogenetic diversity is restricted to any given area) with spatial extent. **d** Spatial distribution of phylogenetic endemism across a global extent. At a global extent, phylogenetic endemism is calculated accounting for the full geographic range of the species. **e** Distribution of phylogenetic endemism (PE) at a regional extent (continent or country). When species ranges span socio-political borders such that phylogenetic endemism is calculated regionally (within a continent or country) without consideration of a species' (or even a clade's) full range, an inflation of phylogenetic endemism results. Sites 1 and 2 refer to the area of circle overlapping with the cells (species ranges) underneath.

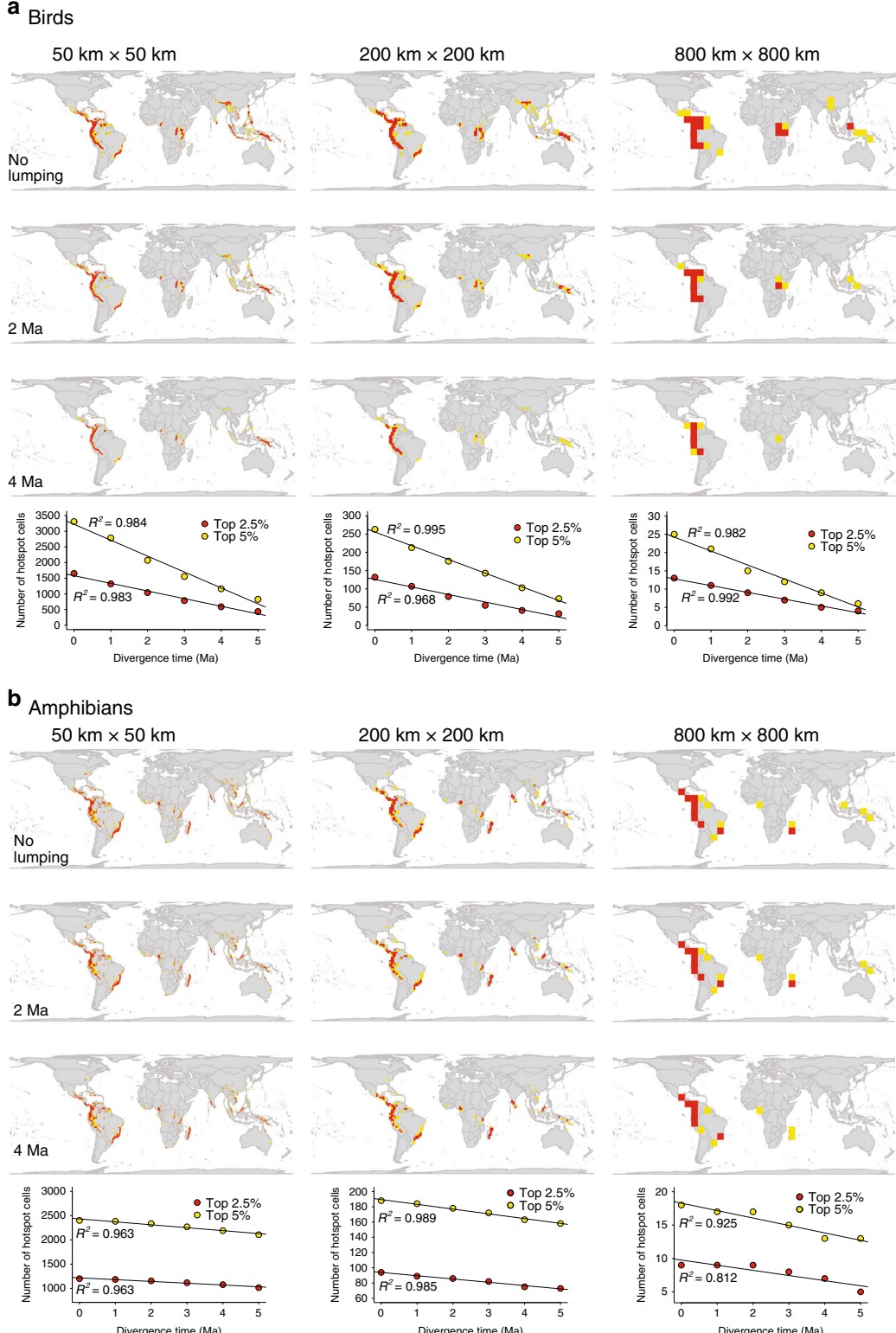

**Fig. 2 Scale-dependent effects of spatial grain and taxonomic treatment on hotspots of weighted endemism, calculated by weighting species according to their range sizes. a** Birds ($n = 10{,}018$ species), and **b** amphibians ($n = 5872$ species). Hotspots were identified using the 2.5% threshold based on the 97.5th percentile values for weighted endemism per grid cell (indicated in red), and 5% corresponding to 95th percentile values (indicated in yellow). Variations of taxonomic treatments in presented results are based on species' divergence times at varying time depths: present-day, from 2 Ma, and from 4 Ma—replicating increasing 'lumping' of taxa. Scatterplots indicate that as taxonomic lumping increases with respect to phylogenetic divergence times, the number of hotspot cells successively decline for both birds (**a**) and amphibians (**b**). Analysis of clade collapse was based on a randomly selected subset of 100 trees from a posterior distribution of 600 trees for birds and 100 trees from a posterior distribution of 10,000 trees for amphibians. The maps are in Behrmann projection. See Supplementary Fig. 2 for the full variation of weighted endemism across grid cells at 50, 100, 200, 400 and 800 km and at varying taxonomic treatment based on species' divergence times from 1, 2, 3, 4 and 5 Ma. Source data are provided as a Source Data file.

and generated new hotspot maps of endemism which we compared with the original data. As species are treated by taxonomic lumping based on their divergence times at varying time depths, our results show that the number of hotspot cells (grid cells with the 97.5th percentile values for weighted endemism), successively decline with increasing spatial grain (Fig. 2), because species lumping collapses smaller ranges into fewer larger parts. This trend of declining weighted endemism across grain sizes was less steep at finer grain sizes (e.g. 50 km) but became more pronounced at coarser grain sizes such as 800 km (Supplementary Fig. 1). This suggests that the more a taxon has been subjected to systematic lumping based on phylogenetic results, the larger is the reduction. This effect highlights a major property of weighted endemism: all species are weighted equally because weighted endemism does not encapsulate phylogenetic relationships[27]. By increasing spatial grain, we may downweigh the effect of true micro-endemics and lose hotspots of endemism in areas such as small oceanic islands or mountain tops[58]. This effect is due to the assumption of larger species ranges and hides key biogeographical processes such as the influence of geographical barriers such as rivers and mountains.

Across taxa, entire avian hotspots of weighted endemism—e.g. Hawaii, Brazil, West Africa, Sri Lanka, Hengduan–Himalaya and Southeast Asia—disappeared at both higher spatial resolutions and under severe taxonomic lumping, i.e. when splits that originated around 2 Ma or longer ago were collapsed (Fig. 2a; see also Supplementary Fig. 2a for full variation of weighted endemism across grain sizes at 50, 100, 200, 400 and 800 km). Similarly, hotspots of amphibian weighted endemism saw great declines at higher grain sizes and under taxonomic lumping, resulting in a greater loss or shrinking of amphibian hotspots that affects geographic regions such as Appalachia and Texas in the US, South Africa, West Africa, Hengduan–Himalaya and Australia (Fig. 2b, Supplementary Fig. 2b). On one hand, coarser grain might capture other evolutionary patterns at large extents, such as allopatric speciation and diversification[7]. On the other hand, while inconsistent taxonomy creates challenges in conservation[40,59], we show here that even if the same taxonomic principle (a standardized cut-off at particular evolutionary depths)[60] is used consistently across the phylogeny, it influences the results. While consistency in species concepts has been advocated in macro-scale studies (e.g. ref. [39]), our results show that using the biological or phylogenetic species concepts can produce different results and might influence conservation prioritization differently.

**Phylogenetic endemism depends on spatial extent**. Hotspots of phylogenetic endemism are influenced strongly by spatial extent, varying along global, continental and local extents at country level (Figs. 1 and 3). Phylogenetic endemism captures the degree to which the phylogeny is restricted to a single area, highlighting the irreplaceability of these areas for the preservation of deep branches of the tree of life[31,61,62]. For both birds and amphibians at the global extent, well-known hotspots in the tropics corresponding to Mesoamerica, the Andes, Africa, Madagascar, Papua New Guinea and South-Central China, plus an additional few in the temperate regions (for amphibians) including Appalachia and the region around Portland Oregon in United States, Southern Chile, Southern Africa and Queensland Australia, emerged as priority regions at fine to intermediate grains but were absent at coarser grain sizes (Fig. 3; Supplementary Fig. 3).

Global patterns of phylogenetic endemism can provide insights into complex evolutionary processes such as dispersal, speciation and extinction shaping large-scale biodiversity patterns[63,64] and may influence the latitudinal diversity gradient, where higher

richness and endemism are observed at lower latitudes for most taxonomic groups[65–67]. However, these effects are lost at the continental to country extents. At the continental extent, for example, hotspots of phylogenetic endemism are less spatially clumped and more dispersed into new locations outside the tropics including southern Europe (Spain, Portugal, Italy, Greece), Georgia, Azerbaijan and Antarctica for birds (Fig. 3a); and Southern Europe, Tasmania and Perth in Australia and New Zealand for amphibians (Fig. 3b). In parallel, some regions that emerged as hotspots at the global extent, including the Atlantic Forest of Brazil, Hawaii, New Zealand and the Oceanic Islands, disappeared at the continental extent (Fig. 3). At the national extent, spatial patterns of phylogenetic endemism became more widespread across countries, clustering more at the socio-political borders of countries and decreasing toward coarser grain sizes (Fig. 3).

Socio-political borders have little ecological relevance because they rarely coincide with ecological boundaries, reflected by the fact that most species ranges span political and continental borders. Patterns of endemism missed in one part of a species range can compromise endemism on either side of the border that the species spans[64]. Biodiversity components on each side of the border are therefore often subject to conflicting management practices[68]. This means that too little or too high attention can be made to border regions for conservation purposes. For example, South Texas in the United States is well-known for its high concentrations of species richness and endemism of birds (including the Green jay *Cyanocorax luxuosus* and the ringed kingfisher *Megaceryle torquata*) and amphibians (e.g., mole and lungless salamanders)[32]. However, most species in this region are mobile and migratory, posing challenges for assessing endemism or extinction risk because their status under conservation legislation can change radically across borders[69,70]. On the other hand, analyses conducted just at the country level can over-estimate endemism levels for species barely reaching into a country (see Fig.1). This is the case for the Red-billed Pigeon (*Patagioenas flavirostris*). This species is widespread in Mexico and central America but has a small-breeding population in southern United States close to Rio Grande in Texas. Unless managers on both sides of a socio-political border adopt compatible management strategies, conservation actions are likely to lead to suboptimal solutions. Thus, the spatial extent of the habitats supporting species should match the scale of management strategies designed to protect the species through international collaborations[71].

The question of which grain is ideal for analysing areas of endemism will depend on the objectives of the study. Rahbek[10] suggests the use of a grain size as small as the smallest range sizes among the species in the study area. Increased grain may reduce biases associated with sampling artefacts because small grain can represent well-known rather than diverse areas. The extent-of-occurrence maps commonly used in biogeographical analyses are drawn by experts to depict the maximum geographical extent of a species and might be compromised by false presences if analysed at too fine a grain. It is therefore generally recommended to interpret these analyses with grain smaller than 1° or 2° latitude/longitude (~110–220 km around the equator) with caution[72–74]. Our results suggest that endemism should be analysed at as high a resolution as the data can allow (which generally is at intermediate grains of 100–200 km).

**Endemism is sensitive to variation in environmental heterogeneity**. To test whether environmental heterogeneity influences patterns of weighted and phylogenetic endemism, we used standard deviation of four commonly used environmental predictors:

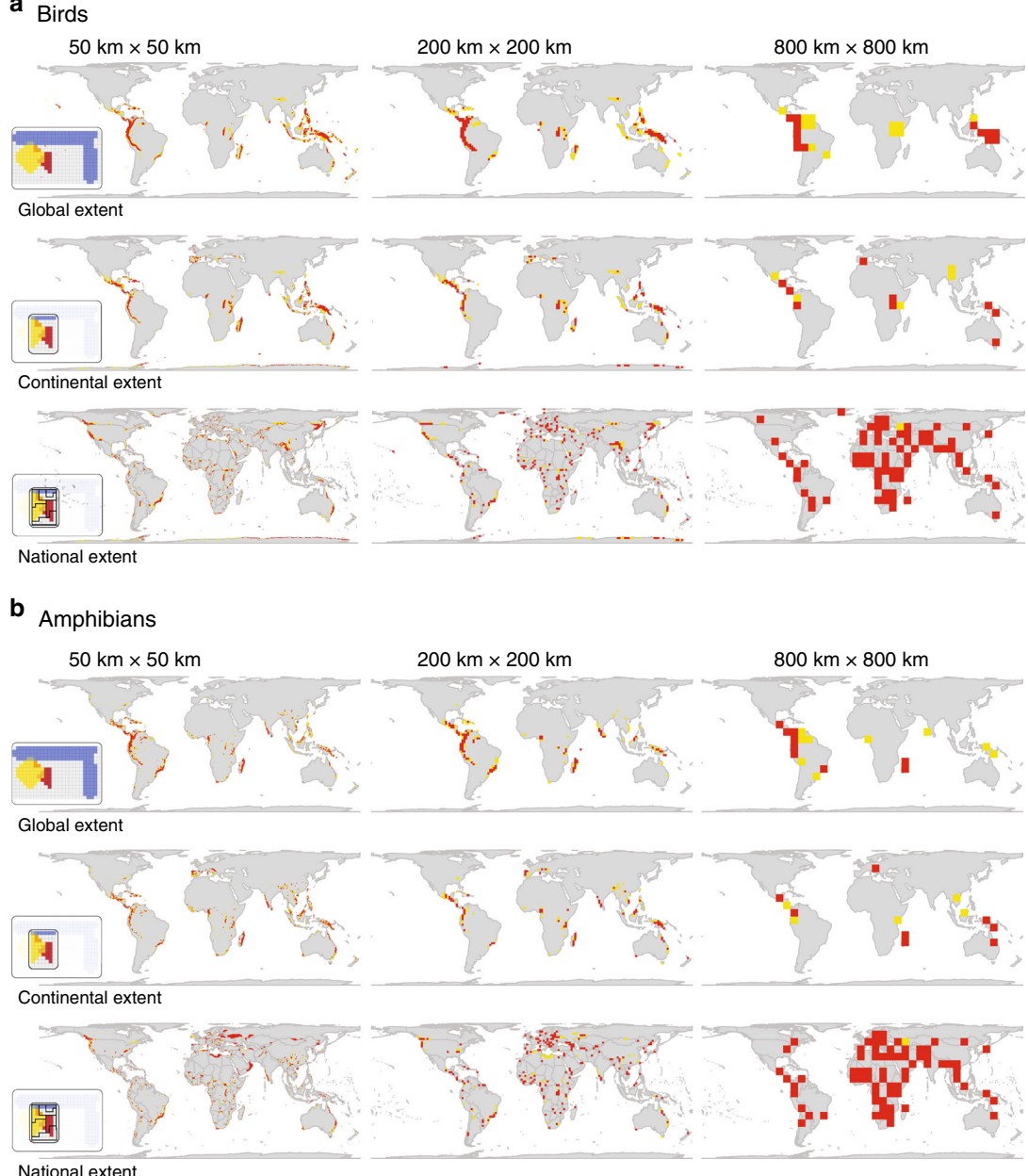

**Fig. 3 Hotspots of phylogenetic endemism (geographic concentrations of phylogenetically and geographically restricted species) are influenced strongly by spatial extent, varying along global, continental and national extents at country level. a** Birds (*n* = 10,018 species), and **b** amphibians (*n* = 5872 species) of the world across three levels of spatial extents (global, continental and national). Global patterns of phylogenetic endemism reflect complex evolutionary processes but are lost at the continental to country extents. When extent is varied, we are varying the scale at which phylogenetic endemism is calculated. Also pictured, on the left in the inset, are simplified visual illustrations of how extent is calculated. Hotspots were identified using the 2.5% threshold based on the 97.5th percentile values for phylogenetic endemism per grid cell (indicated in red), and 5% corresponding to 95th percentile values (indicated in yellow). The maps are in Behrmann projection. See Supplementary Fig. 3 for full variation of phylogenetic endemism across grid cells at 50, 100, 200, 400 and 800 km. Source data are provided as a Source Data file.

elevation (elevation variation henceforth), annual temperature (temperature variation henceforth), annual precipitation (precipitation variation henceforth) and net primary productivity (productivity variation henceforth). We performed these analyses across grain sizes using linear mixed-effects model with a spatial covariate to account for spatial autocorrelation (Fig. 4, Supplementary Tables 1–4). Across clades, our results indicate that, in general, the explanatory power of environmental factors increases with increasing spatial grain for both weighted and phylogenetic endemism and are particularly strong at the two coarsest grains of

400 and 800 km (Fig. 4). For instance, at 800 km, variation (i.e. standard deviation) in precipitation and temperature offer strong predictions of avian weighted endemism (precipitation: beta = 0.32, *p* < 0.001; and temperature: beta = 0.19, *p* = 0.049). The same is true for phylogenetic endemism, which shows strong relationships at intermediate to coarse grains (200–800 km) and is lowest for fine-grained assemblages (Fig. 4), with strong relationships of productivity and precipitation to avian phylogenetic endemism and precipitation to amphibian weighted and phylogenetic endemism. When we compared the relationship between

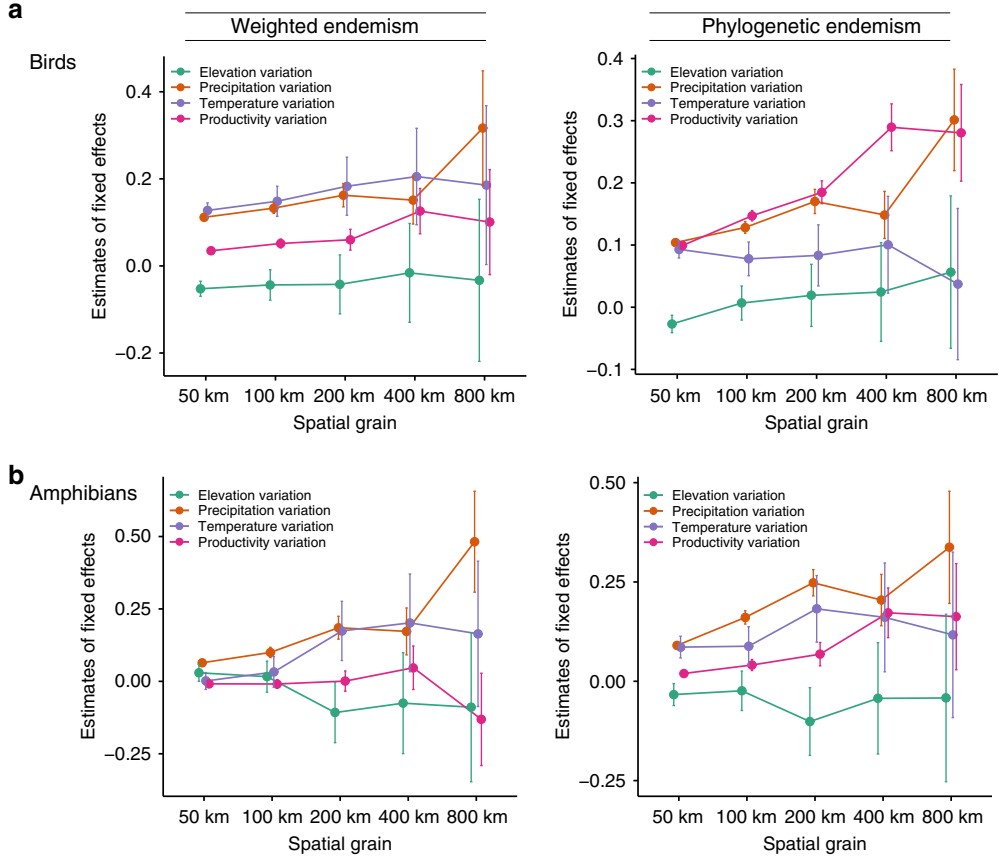

**Fig. 4 The estimates and 95% confidence intervals for the fixed effects predicted by a single linear mixed-effects model of endemism with all predictors (temperature variation, precipitation variation, productivity variation, elevation variation, spatial autocovariate and random effects of continent identities for each grid cell) across grain sizes. a** Birds and **b** amphibians. These models indicate that our findings are explained by environmental heterogeneity at coarser grains and to a lesser extent at finer resolutions (see Supplementary Tables 1–4). Significance was assessed by comparing likelihoods of the fitted objects. Source data are provided as a Source Data file.

endemism and landscape heterogeneity (measured as heterogeneity of altitude above sea level) versus latitude, we found that the residuals of landscape heterogeneity and endemism are concentrated in the tropics and decline with latitude (Supplementary Fig. 4). This trend is particularly strong at coarser grain size. Patterns of endemism might therefore have been erased in regions with highly seasonal climates, corroborating the high endemism in the tropics, and mainly in areas under the climatic influence of thermally stable tropical oceans[75].

Environmental heterogeneity is assumed to promote dispersal barriers that may decrease species diversity leading to increased speciation rates[7]. As grain size increases, climatic variables are often assigned a summary value for the grid cell (which can be the centroid value, mean or median), and can directly bias the importance of regional climate heterogeneity or the locality from where species actually occur, thus leading to spurious conclusions[76,77]. Our results highlight the limitation of comparing endemism–environment relationships at single grains, such that our approach can help in locating hotspots of endemism that are more meaningfully associated with the environmental features of the region. Thus, a multi-grain approach to endemism–environment relationships should be considered in model testing and conservation planning.

**In situ conservation varies with scale.** To highlight the critical gaps in protecting areas of endemism across scales, we mapped hotspots of weighted and phylogenetic endemism for birds and

amphibians of the world across grain sizes, spatial extents and taxonomic treatments. We then assessed the scale to which areas of endemism are captured in at least 10% of the current network of protected areas. The 10% threshold is a conservative target of coverage by protected areas advocated for safeguarding biodiversity[78,79]. Overall, we reveal that only 22–29% of avian endemism hotspots, and 24–25% of amphibian hotspots, meet a minimum target of merely 10% potential coverage by the global system of protected areas (Fig. 5). Across scales, the situation is even more alarming. Hotspots of weighted endemism for both birds and amphibians are more protected at finer to intermediate grain sizes (50–200 km), with up to 28–33% coverage by protected areas for birds (Fig. 5a) and 26% for amphibians (Fig. 5b). This pattern is insensitive to the extent of taxonomic over-lumping. Importantly, all hotspots of phylogenetic endemism—regardless of grain size or spatial extent—fall below the critical 10% coverage target for protection, with the exception of hotspots of amphibian phylogenetic endemism analysed at the continental extent that meet the minimum protection threshold of 10% by protected areas (Fig. 5). Overall, we find widespread deficits of protection for endemism hotspots regardless of grain size, spatial extent or taxonomic treatment.

In conclusion, our study shows that changes in taxonomic treatments, spatial grain and spatial extent strongly influence patterns of endemism. Taxonomic lumping can be detrimental to conservation if species are delisted as a result, with subsequent cessation in monitoring and policy efforts for their protection[59]. Conversely, splitting can lead to suboptimal conservation

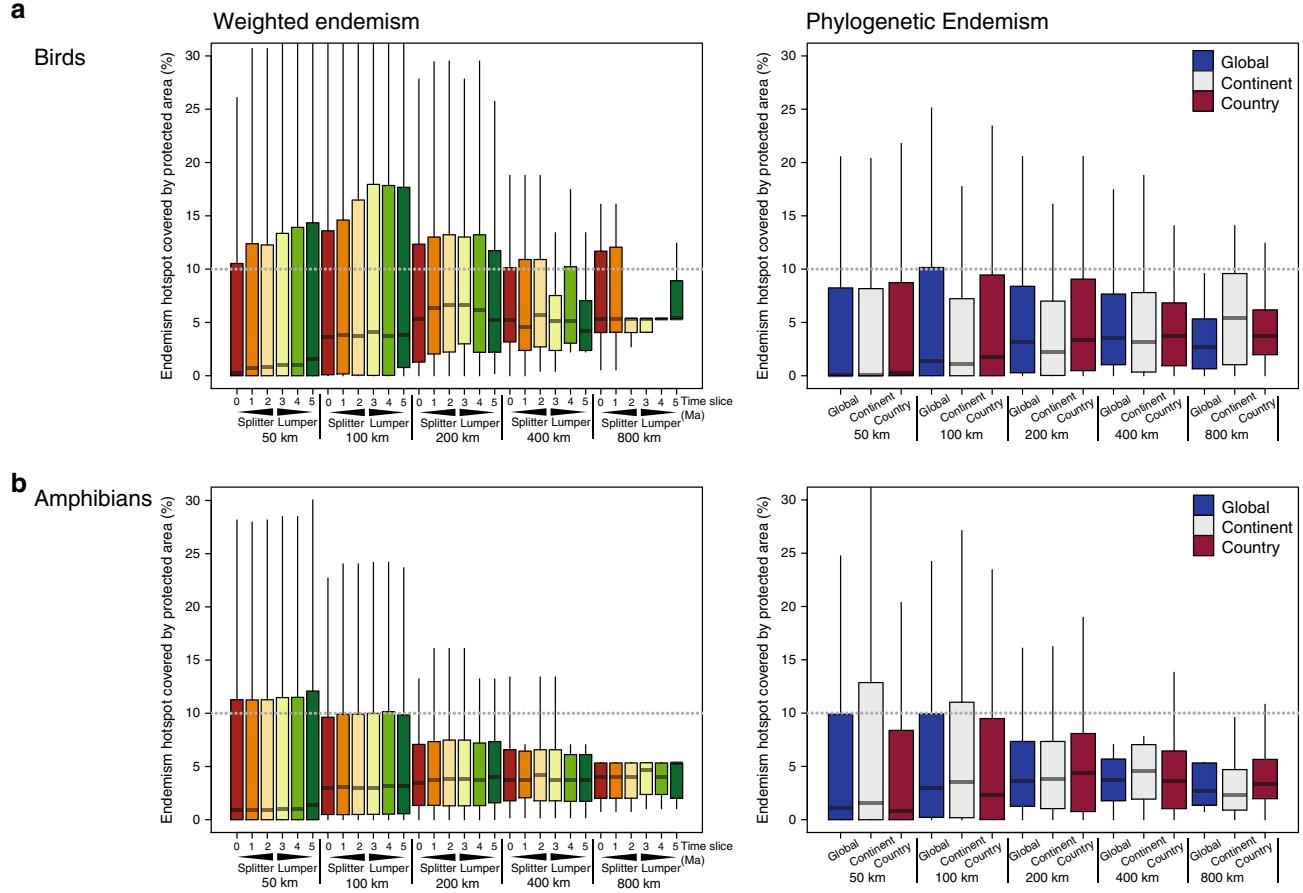

**Fig. 5 Relationship between variation in grain size and the proportion of endemism hotspots covered by the global systems of protected areas. a** Bird (*n* = 10,018 species) weighted endemism (left) and phylogenetic endemism (right), and **b** amphibian (*n* = 5872 species) weighted endemism (left) and phylogenetic endemism (right). Weighted endemism was calculated by weighting species according to their range sizes, whereas phylogenetic endemism weights phylogenetic branches by the inverse of geographical range size of that branch, such that the branch lengths of range-restricted species are given more weight. The dotted lines represent the 10% threshold corresponding to the minimum representation target for sustaining species persistence. These findings demonstrate widespread deficits of protection for endemism hotspots regardless of grain size, spatial extent or taxonomic treatment. The bottom and top of boxes show the first and third quartiles respectively, the median is indicated by the horizontal line, the range of the data by the whiskers. Source data are provided as a Source Data file.

solutions and management problems because populations are managed as distinct units without any assisted gene flow, potentially causing inbreeding issues. Splitting can also lead to potentially incorrect spatial conservation prioritization, which may limit the number of biological species that are properly managed. Coarse spatial grain misinterpretations can overlook important areas of endemism which may become vulnerable and degraded due to anthropogenic disturbance. Endemism represents one important but overlooked component in conservation[47,48,80]. Our study highlights the large impact that scale can have on understanding biodiversity patterns and prioritising areas for conservation and stresses the importance of analyses that are robust to assumptions on taxonomy, spatial grain and extent.

## Methods

**Species distribution data.** The geographic distributional data for birds were obtained from BirdLife International[81], a comprehensive global geographic database for all land and non-pelagic species (*n* = 10,079 species) available as range map polygons. Range maps for all amphibians were obtained from the IUCN Red List database (https://www.iucnredlist.org/resources/spatial-data-download) of the native extent-of-occurrence of all amphibian species (*n* = 6337 species). Both sets of maps represent the extent-of-occurrence of the breeding ranges based on museum specimens and direct field observations which have been validated by experts. We matched the range maps to standardized taxonomic authorities including Frost[82] and data from the American Museum of Natural History

(AMNH; http://research.amnh.org/vz/herpetology/amphibia/index.php) for amphibians and ref. [38] for birds.

**Phylogenetic data.** Phylogenetic data for birds comprised a phylogeny for all extant bird species, representing 10,079 species, which was based on a distribution of 10,000 possible tree topologies from ref. [55]. The amphibian phylogeny comprised a phylogeny of 7238 species (94% of all extant amphibians) based on 15 genes on a distribution of 10,000 possible tree topologies from ref. [83]. To account for phylogenetic uncertainty in our analyses for both birds and amphibians, we drew 100 trees at random from a posterior distribution of fully resolved trees generated in ref. [55] for birds and ref. [83] for amphibians.

**Degree of in situ protection.** We quantified the extent to which the global network of protected areas covers hotspots of endemism across grain sizes, spatial extent and taxonomic treatment using the World Database on protected areas (http://protectedplanet.net/)[84]. Our analysis was done on the basis of all terrestrial protected areas classified as IUCN categories I–VI as having sufficient protection status that increases the likelihood that species are well protected. For each hotspot cell, we quantified the amount of polygon area and examined the proportion of cell overlapping with global system of protected areas. We adopted a 10% cut-off spatial coverage by protected areas corresponding to a conservative coverage target for effective biodiversity protection[78,79].

**Data analysis.** We constructed a binary presence–absence matrix by overlapping the extent-of-occurrence range map of each bird species with equal-area grid cells using the *polys2comm* function in our new R package phyloregion[60]. These grid cells were mapped at five consecutive grain sizes following the Behrmann

equal area projection system: $50 \times 50$, $100 \times 100$, $200 \times 200$, $400 \times 400$ km and $800 \times 800$ km$^2$. At each grain size, we calculated species weighted and phylogenetic endemism.

We used a variant of Laffan and Crisp's[28] weighted endemism metric, defined as the sum of the number of species present in each cell in a local neighbourhood, weighting each by the fraction of the area they inhabit[28]. Weighted endemism (WE) is expressed as:

$$WE = \sum_{\{t \in T\}} \frac{r_t}{R_t},$$

where $R_t$ represents the full geographic range of taxon $t$, and $r_t$ is the local range of taxon $t$, with the range of a taxon counted in units of number of grid cells in which it is found. We estimated changing spatial grain in weighted endemism for both birds and amphibians under two scenarios of taxonomic treatment: splitting and lumping. We quantified taxonomic lumping, by successively slicing the phylogenetic tree at various time depths (from 1, 2, 3, to 5 million years ago (Ma)), collapsed nodes and ranges that originated at each time depth (using phyloregion's function collapse_range), and generated new maps of endemism. It is not possible based on available data to investigate the effects of increased splitting on endemism patterns, but it is feasible to assume that some of the hotspots we identify as sensitive to taxonomy may be so in both directions. We used the function get_clades, also in our new R package phyloregion[60], to manipulate the phylogenetic tree and collapse nodes and ranges at varying time depths. Weighted endemism was calculated using the function weighted_endemism(x), also in our new R package phyloregion[60], where $x$ is a community matrix or data frame. Our results were integrated across variations of tree topologies and branch lengths for both birds and amphibians by repeating the weighted endemism calculation for each 100 trees from the posterior distribution of trees and taking the median across grid cells.

Phylogenetic endemism was measured as the total phylogenetic branch length spanned by species in an area, dividing each branch length by the global range size of its descendant clade measured in Myr/km$^2$ ref. [29]. Phylogenetic endemism was calculated using the function phylo_endemism(x, phy, weighted = TRUE) in our R package phyloregion[60], where $x$ is a community matrix or data frame, and phy is a phylogenetic tree object. Phylogenetic endemism (PE) is expressed as follows:

$$PE = \sum_{\{i \in I\}} \frac{L_i}{R_i},$$

where $\{I\}$ represents the set of branches connecting species to the root of a phylogenetic tree, $L_i$ is the length of branch $i$, expressed as proportion of the total length of the tree and $R_i$ is the range size of the clade. Because we assumed that phylogenetic endemism would vary strongly with spatial extent, we varied the phylogenetic endemism analysis across successive spatial extents (global, continental and national) and mapped the hotspots at each spatial extent (see explanatory Fig. 1). At the continental or country extents, phylogenetic endemism was calculated based only on the species present in that particular inference space by placing cells in continents or countries based on centroids. When phylogenetic endemism is calculated at too narrow a scale, anomalous results may appear. However, insights into more complex evolutionary processes such as dispersal, speciation and extinction shaping biodiversity patterns are captured at larger continental to global extents (e.g. ref. [52]). We integrated our results across variations of tree topologies and branch lengths for both birds and amphibians by repeating the phylogenetic endemism calculation for each 100 trees from the posterior distribution and taking the median across grid cells for further analysis.

We chose biodiversity hotspots as the basis for quantifying scale dependence of endemism because hotspots can guide allocation of limited conservation resources (e.g., ref. [85]) and endemism lies at the core of understanding the variation of biodiversity across space[47]. Hotspots are areas with significant species richness and endemism. Hotspots for each metric were defined as the 2.5% of grid cells with the highest values of weighted and phylogenetic endemism[48,80]. To assess uncertainty in the results, we re-ran all analyses by increasing the threshold percentage to 5% to examine if a different threshold percentage value altered the areas identified as hotspots in our analyses. Hotspots were calculated using the function hotspots(x, prob =; 2.5) in our new R package phyloregion[60], where $x$ is a vector on which to compute hotspots analysis, and prob the threshold quantile for representing the highest proportion of biodiversity in an area. By default, the threshold is set to prob = 2.5%.

**Environmental data and heterogeneity**. We selected key environmental factors that are commonly used to examine biodiversity-environment associations. These variables included mean annual temperature, mean annual precipitation, annual net primary productivity and elevation. Mean annual temperature, mean annual precipitation and elevation were downloaded as raster layers from the WorldClim database[86] at a resolution of 2.5'. Annual net primary productivity was downloaded from NASA Moderate Resolution Imaging Spectroradiometer at a resolution of 1 km and calculated using the MOD17 algorithm. These variables were converted to Behrmann equal-area projection using the function projectRaster in the R package raster[87].

We defined 'environmental heterogeneity' as the variation of environmental factors in each cell, obtained by taking the standard deviation of the environmental variables (temperature, precipitation, elevation and productivity) in each cell and across grain sizes (50, 100, 200, 400 and 800 km). Henceforth, we refer to the standard deviation of these four environmental variables i.e., mean annual temperature, mean annual precipitation, annual net primary productivity and elevation, as temperature variation, precipitation variation, productivity variation and elevation variation, respectively.

**Linear mixed-effects model**. We fit a single linear mixed-effects model to analyse the effect of environmental heterogeneity on patterns of endemism (PE or WE) across grain sizes. A linear mixed-effects model allows the modelling of data as a combination of fixed effects, random effects and independent random error, and are especially useful when there is non-independence in the data[88]. The standard form of a linear mixed-effects model is expressed as:

$$Y_i = x_i \beta + s_i + \varepsilon_i,$$

where $Y_i$ represents the response variable at grid cell or location $i$, $x_i$ is a matrix of the observations (explanatory variables) used as predictors (covariates), and $\beta$ is a vector of the unknown regression coefficients, which are often called fixed effect coefficients, $s_i$ is a matrix, similar to $x$ that captures the complex covariance structure for spatial autocorrelation and $\varepsilon_i$ is the random measurement error (residuals). All explanatory variables were standardised prior to statistical analyses so that all variables had a standard deviation of 1 and a mean of 0. This ensures that the estimated coefficients are all on the same scale and for easier comparison of effect sizes.

The linear mixed-effects model was fit as a single model with all the variables in one model predicting endemism (PE or WE) as a function of the four environmental variables (temperature variation, precipitation variation, elevation variation and productivity variation), with continent identity of grid cells as a random effect, allowing us to include any idiosyncratic differences between continents. The model also includes a spatial covariate of geographical coordinates as an additional predictor variable to account for spatial autocorrelation. The spatial covariate was created as a matrix of the coordinates of each cell's centroid corresponding to the geographical Cartesian $x/y^-$ coordinates (longitude and latitude), and was calculated with the function autocov_dist in the R package spdep[89]. For each focal cell, we varied the weighting function and neighbourhood sizes using the next one to two cell neighbours to remove spatial autocorrelation (function nb2listw in spdep[89]). We used the maximum likelihood optimization criterion over restricted maximum likelihood, to allow significance testing via model comparison.

The linear mixed-effects model was fit in R v.3.6.3[90] with the lme function in the R package nlme[91]. The variations of environmental variables on endemism are presented as estimated coefficients of the fixed effects and their 95% confidence intervals in the R package nlme[91]. A vignette, with a worked example, data and R codes describing all the steps for the analyses, is also provided in our R package's website (https://darunabas.github.io/phyloregion/articles/heterogeneity.html).

**Reporting summary**. Further information on research design is available in the Nature Research Reporting Summary linked to this article.

## Data availability
All data necessary to repeat the analyses described here have been made available through the Dryad digital data repository (https://doi.org/10.5061/dryad.wh70rxwhs)[92]. The source data underlying Figs. 2–5 are provided as a Source Data file.

## Code availability
All scripts and code necessary to repeat the analyses described here have been made available in the new R package phyloregion[60].

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

## Acknowledgements

We thank Texas A&M University-Corpus Christi for logistic and financial support. B.H.D. is supported by start-up funds from Texas A&M University-Corpus Christi, S.F. is supported by the Swedish Research Council (#2017-03862), and A.A. is supported by the Knut and Alice Wallenberg Foundation, the Swedish Research Council, the Swedish Foundation for Strategic Research, and the Royal Botanic Gardens, Kew. We are grateful to Rhian J. Smith for help with language editing.

## Author contributions

B.H.D. designed the study and ran the analyses with help from S.F., H.F and A.A. B.H.D. wrote the paper with substantial contributions from all co-authors.

## Competing interests

The authors declare no competing interests.
