## [Peer Review File · Nature Communications]

Reviewers' Comments:

Reviewer #1:

Remarks to the Author:

The manuscript "Endemism patterns are scale dependent" has a potential. Although it is more an evolution of the current concepts, rather than a revolution, it is sufficiently multi-faceted to come across as novel and thorough on a level that I would expect for Nature Communications, although I do have some reservations below. It does empirically corroborate the (broadly expected) scale-dependency of global patterns using two global datasets, and to my knowledge this has not yet been done. Another novelty is the step from a simple endemism-area relationships to spatial patterns; there is some literature on the former, but almost nothing for the latter. Further, there is the added value of looking at scale-dependence of patterns of phylogenetic endemism (not done yet) and relating this to protection in reserves (not done yet); these are welcome additions. Since endemism is such an important quantity in conservation biogeography, I predict that the paper will attract citations.

The bottom line is: I like the paper, it seems mostly solid and relevant for the field, and the reported patterns have not yet been empirically demonstrated. It also feels like something that is above the standard of more specialized ecological journals, and thus Nature Communications seems like a suitable venue for this.

I do have some comments mostly about clarity of the presentation, but all of them should be straightforward to address.

MAJOR COMMENTS

While reading the paper, I was confused by the terminology. My main concern is about how grain and extent are often made synonymous with scale (e.g. lines 108, 144, 156, 164, 191, 193, 432, and elsewhere). I strongly suggest to use ctrl+H function to locate all instances of the word "scale" and replace it with the specific terms "grain" and "extent" wherever possible. I would even think about doing this in the title, but there it is perhaps a matter of taste.

I miss a conceptual figure for the main text that would precede all the results. Something that would illustrate all the key concepts and/or hypotheses, and provided some graphic handle to them. Perhaps something derived from Fig. M1 (which I like a lot)? I think that that would also clarify some of the terminological confusion. Alternatively, add a little illustrative panel to each of the main text figures. Anything that makes the ideas more graphic, and that also makes the ms more memorable for those who only skim through it.

I am confused by Figure 1 and 2, and how the patterns in these are described in the main text. First paragraphs of the Results & Discussion mention "declines of endemism" (lines 114, 132), but the figures only show maps from which it is unclear what actually declines (number of cells? their total area? their proportion? average endemism value?), and if it is a statistically significant decline. Maybe try to be more exact in the wording, and support the statement with a statistical test or additional scatterplot figure. Second, it is unclear how extent works in Figure 2, since all the maps have an identical extent, but they do differ in their grain (making them easy to confuse with Fig. 1). Perhaps this is just a communication issue, but could you try to add some visual clue to how extent plays a role in this Figure 2? Maybe also improve the labels to, e.g., state "Global extent" instead of just "Global"? Overall, I think that this figure, and the associated paragraph, are really confusing in terms of scale/extent/terms, and what varies where. And maybe part of Figure M1 can make it to Figures 1 and 2, so that each empirical pattern is accompanied by a little conceptual cartoon that illustrates it? These are all just suggestions, up to you how you deal with this in order to make all clearer.

Figure 3. I have several issues here, and I hope that I am misunderstanding something, because if I am not, the results might be a bit different from what the authors state. Please double check.

Specifically:

- First, I don't see how the effect is "overwhelming" at fine scales (as described in caption and in the main text). It actually seems closer to zero than at coarse scales, but I may be missing something.
- Second, and forgive me if I'm wrong, but is the y-axis really log endemism? Isn't it some sort of effect size? Guessing from the range of values, it seems like these are standardized model coefficients (but if this is the case, then the effects are mostly quite weak).
- Third, some points have no confidence intervals. Is this because they are narrower than the point size, or is there an error?
- Fourth, you might want to add a gray or dashed horizontal line cutting through zero (if y-axis is effect size).

MINOR COMMENTS

Line 30: I don't think that diversification processes belong to the abstract. They are not what you directly study.

Figures 3, 4, 5 and maybe other figures (and tables) - I would add what "weighted" means, i.e. what it is weighted by. This is because figures and their captions should be self-explanatory, understandable without the need to the main text. Many will jump directly to the figures, skipping the main text.

While reading your paper, I did a little search for literature, just to see what is out there, and I skimmed through some of my papers and the citations within. The main conclusion is that your contribution does indeed seem new. However, I thought that maybe it could be useful/inspirational to give you some other references that seem relevant - here it's completely up to you whether to include them or not (by all means avoid "including citations to please the reviewer" situation).

- Ochoa-Ochoa 2014 Biol. J. Linn. S., 10.1111/bij.12201 deal with spatial patterns of endemism, and their scale-dependency.
- IUCN red list guidelines, <https://cmsdocs.s3.amazonaws.com/RedListGuidelines.pdf>, Figure 4.6 and related sections, and citations therein. I think that this is the most practical and applied case where the scale-dependency of range size is used. Since range size is the key ingredient in determining endemism, it seems highly relevant. For a more comprehensive treatment of the grain-dependency of range size and extinction risk see the many works by William Kunin.
- Townsend-Peterson & Watson 1998 Div. Distr. 10.1046/j.1472-4642.1998.00021.x. This seems like something that you might want to check/cite, or see who cites it.
- Keil et al. 2015 Nature Communications, 10.1038/ncomms9837. Their Figure 3 shows how endemism and phylogenetic endemism scales with area (i.e. the endemic-area curve, EAR, and phylogenetic endemism area curve, PDXAR), and how taxonomic and phylogenetic endemism are related. Their Figure 4 shows how this scaling varies across continents.
- Graham et al. 2018 GEB 10.1111/geb.12686 seems like something that you should definitely look at and cite it, since it is tightly related to your "taxonomic lumping".
- Storch & Sizing 2008 Folia Geobotanica 10.1007/s12224-008-9015-8 deal with similar issues.
- I am not Storch.

Fig. 4 – State what the box and whiskers show.

Fig. M1 – Replace the "ANALYSIS" label in the figure by either extent, or grain, whatever is appropriate. Also, try to think about how to migrate a version of such figure to the main text that

would illustrate all your important concepts and scalings.

Lines 147 – 153: The sentence is too long and complicated.

Lines 203 – 206. Please state that you used standard deviation of these variables, not their mean values.

I really appreciate that the code and data are publicly available. Great job!

Reviewer #2:

Remarks to the Author:

This MS covers an interesting issue, of quite some interest for how to use biodiversity data to prioritise conservation actions. It analyses large phylogenies in combination with large online distribution databases. The question of scale-dependence and species level taxonomic has been covered in several other papers, but this MS represents an important extension, since it is done for the whole world, and it presents an objective measure of degree of systematic 'lumping'. Here, Fig. S2 is very illustrative. However, the MS also contains some very strange results, which require better explanation and interpretation. The Discussion seems to suffer from an insufficient understanding of the global variation in species diversity.

The maps of phylogenetic endemism (Fig. 2 and S3) initially did not make sense to me, especially the category 'national', which shows red grid-cells in areas that only have few and widespread species (especially in the Western Palearctic, north of the Alps; note that Europe has only two Amphibia species with restricted distributions outside the Mediterranean Area). When I came to the Data Analysis section I finally began to understand what is the problem, and I need to go a bit into detail here to explain my interpretation of this seemingly odd result: Some of these grid-cells are in areas with only two or three widespread amphibian species, so this is a very small sample, and I think that these areas were recognized as 'areas of phylogenetic endemism' because they combine wide species distributions and phylogenetically highly overdispersed communities. Birds are less extreme, but European species (outside the Mediterranean area) are generally widespread, but again, the communities are generally phylogenetically overdispersed, leading to some grid-cells being detected as areas of high 'phylogenetic endemism'. Note that Europe N of the Alps is generally characterized by widespread and phylogenetically independent species re-colonizing the area over and over again (during interglacial periods) from Pleistocene refuges in the South and in Asia (the only exception is *Loxia scotica*, of doubtful species rank) (quite unlike in the Eastern Palearctic, which mainly remained ice-free during the Pleistocene, and therefore was an area of continued speciation. This general pattern, with predominance of old species in the Western Palearctic (and in some tropical and austral parts of the world), is illustrated for Passerine birds in Chinese Birds 4 (2013): 132-43.

In order to avoid confusion over how these strange geographical patterns appear, I think, first of all, that the Introduction should provide a more transparent explanation of what the 'phylogenetic endemism' means (line 54). You could simply move the explanation on lines 520-522 to the Introduction. Maybe even with an additional statement, that this means that even areas with quite widespread species are detected as areas of phylogenetic endemism if the local communities are phylogenetically highly overdispersed. You could also remove the problem with too small samples by removing areas with very few (for instance <10) species.

The 'country level' aspect I find quite odd. Most countries like to flag, in their national biodiversity reviews, their list of endemic species (recorded in no other nation), and it could be interesting to see to what extent this would be affected by taxonomy. But I fail to see the relevance of analyzing the occurrence of small marginal populations within a national territory of species that are widespread,

common and in the 'lower risk category' outside that national territory. So what is the relevance of emphasizing (line 278) of the tiny population of ringed kingfisher in Texas, when this species is distributed almost all over Central and South America? IUCN has had great problems with agreements over criteria for national red-lists for species that are not globally threatened, and I don't think that the inclusion of national concerns over marginal populations adds much to an otherwise interesting analysis.

Lines 203-226: I don't think there is much novel in this analysis. It is well established that landscape heterogeneity is an important factor explaining differentiation of local populations and thereby the degree of endemism. A particularly strong demonstration of this is by Rahbek et al. in *Science* 365 (2019): 1108-1113. But it is important to note that the pattern is erased in regions with highly seasonal climates, which explains why high endemism is so strongly concentrated to the tropics, and mainly to areas under the climatic influence of thermally stable tropical oceans. If the authors could demonstrate how correlation between endemism and landscape heterogeneity depends on latitude, that would be a novel element.

Other relevant papers would be by Rahbek et al. in *Proc. Royal Soc. B* 274(2007): 165-175 regarding the analysis of scale-dependence, and Dillon & Fjeldså in *Ecography* 28 (2005): 682-692 regarding significance of how species are defined.

Details: In line 25, I don't think the problem is 'poorly known', but it has not been analyzed in a systematic way. So maybe use the word 'analyzed' instead of 'known'.

Reviewer #3:

Remarks to the Author:

The paper by Daru and collaborators investigate the relationship between endemism patterns (Weighted Endemism and Phylogenetic Endemism) and spatial scale (i.e. grain size). I think the manuscript is well written and the methods and analyses are sound. This is a stimulating topic and should be of interest to a broad audience of readers of *Nature Communications*. I only have a few concerns/comments that I think the authors need to clarify. I think these editions would improve the manuscript.

Line 52-66 – I think these issues are too methodological. The previous paragraph does a good job explaining that all biodiversity patterns are scale-dependent and that different mechanisms may operate on different scales. I think at the beginning of the Introduction the authors should better explore the broader context of what are the potential mechanisms that might explain variation in endemism across scales.

Line 67-69 – It is not clear why PE "offers a potential solution to deal with new taxonomic knowledge in conservation strategies because some lineages and areas harbor far more endemic diversity than would be expected from species ranges alone"

Line 69-82 – I also think that this paragraph should bring a more general account of what processes could lead to the patterns seen in PE at different scales

Line 88-92 – These predictions are not very clear to me. In the amphibian example, you mentioned that the effect of scale on endemism should be strongest at finer grains. Isn't the scale effect necessarily a change across scales?

Line 93-95 – I would argue that this is not a very interesting way of framing your main hypothesis and therefore the main goal of your article. As you stated in the first paragraph all biodiversity patterns are influenced by spatial scale.

Line 100-104 – I think these are interesting questions but the previous section of the introduction did not properly provide the necessary contextualization related to these questions (except for some text regarding question i). It would be better to have a better contextualization of these topics and some predictions.

Line 113-115 – Perhaps 'richest' is not the best word here since it could confuse the reader. From the figure, it is difficult to see what the authors refer to. Perhaps a table containing the number of grid cells in the high 2.5% could help. Also, as the cell size increases the number of total cells decreases so, the decrease in the number of cells with highest 2.5% WE could be proportional to this total number of cells decrease?

Line 121-123 – Is this what the data shows? Did you observe a loss of hotspots of endemism in islands and mountain tops?

Line 127-131 – This answered my question above so I think you should combine this somehow.

Line 139-141 – Is there anything more you can say about the taxonomic issue? It seems trivial that using different concepts would produce different results.

Line 148 – I suggest replacing the term 'biodiversity hotspot' here. This can confuse the reader since is a well-know term in other literature.

Line 155-158 – This sentence is too generic it would be more helpful to state how PE can influence on those things.

Line 203-206 – Earlier in the text the authors need to provide a succinct account on how these environmental variables are expected to influence endemism and how they should change with scale.

Reviewers' comments:

Reviewer #1 (Remarks to the Author):

The manuscript “Endemism patterns are scale dependent” has a potential. Although it is more an evolution of the current concepts, rather than a revolution, it is sufficiently multi-faceted to come across as novel and thorough on a level that I would expect for Nature Communications, although I do have some reservations below. It does empirically corroborate the (broadly expected) scale-dependency of global patterns using two global datasets, and to my knowledge this has not yet been done. Another novelty is the step from a simple endemism-area relationships to spatial patterns; there is some literature on the former, but almost nothing for the latter. Further, there is the added value of looking at scale-dependence of patterns of phylogenetic endemism (not done yet) and relating this to protection in reserves (not done yet); these are welcome additions. Since endemism is such an important quantity in conservation biogeography, I predict that the paper will attract citations.

The bottom line is: I like the paper, it seems mostly solid and relevant for the field, and the reported patterns have not yet been empirically demonstrated. It also feels like something that is above the standard of more specialized ecological journals, and thus Nature Communications seems like a suitable venue for this.

I do have some comments mostly about clarity of the presentation, but all of them should be straightforward to address.

RESPONSE

We thank the reviewer for the positive remarks. We hope our edits, described below, have addressed the reviewer's concerns.

MAJOR COMMENTS

While reading the paper, I was confused by the terminology. My main concern is about how grain and extent are often made synonymous with scale (e.g. lines 108, 144, 156, 164, 191, 193, 432, and elsewhere). I strongly suggest to use ctrl+H function to locate all instances of the word “scale” and replace it with the specific terms “grain” and “extent” wherever possible. I would even think about doing this in the title, but there it is perhaps a matter of taste.

RESPONSE

The reviewer raised the important point of using consistent terminology and suggested replacing “scale” with “grain” or “extent” where possible. We appreciate this suggestion and have revised our usage of the terms “grain” or “extent” to be more specific throughout the manuscript where necessary. We have also provided a definition for scale in the Introduction: “*Our definition of scale encompasses three components: grain, extent and taxonomic treatment*”. However, there are few instances where we retain our use of “scale” e.g. in the title and Lines 124–126, because it is more an umbrella term that encapsulates the variation of endemism with grain, extent and taxonomic treatment³, which is the main focus of the paper.

I miss a conceptual figure for the main text that would precede all the results. Something that would illustrate all the key concepts and/or hypotheses, and provided some graphic handle to them. Perhaps something derived from Fig. M1 (which I like a lot)? I think that that would also clarify some of the terminological confusion. Alternatively, add a little illustrative panel to each of the main text figures. Anything that makes the ideas more graphic, and that also makes the ms more memorable for those who only skim through it.

RESPONSE

We think this is a very good point, and indeed would strengthen our main message. We now provide a conceptual figure illustrating the scale-dependence of weighted endemism (WE) and phylogenetic endemism (PE) in a new **Fig. 1**. Here, we illustrate the importance of taxonomic treatment on WE and extent on PE.

Weighted endemism

Phylogenetic endemism

Fig. 1 | Schematic of the variation of endemism with spatial extent, grain size and taxonomic treatment. **a** and **b**, Changes in weighted endemism (species richness inversely weighted by species ranges) under a scenario of taxonomic lumping at divergence time of 1 million years ago (Ma). As the node and ranges that originated at 1 Ma were collapsed (**b**), new hotspot maps of weighted endemism were generated which we compared to the original data (**a**). **c** Variation of phylogenetic endemism (the degree to which phylogenetic diversity is restricted to any given area) with spatial extent. **d** Spatial distribution of phylogenetic endemism across a global extent. At a global extent, PE is calculated accounting for the full geographic range of the species. **e** Distribution of phylogenetic endemism (PE) at a regional extent (continent or country). When species ranges span socio-political borders such that PE is calculated regionally (within a continent or country) without consideration of a species' (or even a clade's) full range, an inflation of phylogenetic endemism results.

I am confused by Figure 1 and 2, and how the patterns in these are described in the main text. First paragraphs of the Results & Discussion mention “declines of endemism” (lines 114, 132), but the figures only show maps from which it is unclear what actually declines (number of cells? their total area? their proportion? average endemism value?), and if it is a statistically significant decline. Maybe try to be more exact in the wording, and support the statement with a statistical test or additional scatterplot figure. Second, it is unclear how extent works in Figure 2, since all the maps have an identical extent, but they do differ in their grain (making them easy to confuse with Fig. 1). Perhaps this is just a communication issue, but could you try to add some visual clue to how extent plays a role in this Figure 2? Maybe also improve the labels to, e.g., state “Global extent” instead of just “Global”? Overall, I think that this figure, and the associated paragraph, are really confusing in terms of scale/extent/terms, and what varies where. And maybe part of Figure M1 can make it to Figures 1 and 2, so that each empirical pattern is accompanied by a little conceptual cartoon that illustrates it? These are all just suggestions, up to you how you deal with this in order to make all clearer.

RESPONSE

We appreciate the reviewer's comments on our previous figures 1 and 2, which we have now revised and renamed to **Figs. 2** and **3**, respectively. We have generated a new figure that replaces the previous Fig. 1 (now **Fig. 2**) by incorporating scatterplots along with R-squared values showing the number of hotspots cells against taxonomic lumping in terms of divergence times (see below). Here, the negative slopes in the scatterplot suggest that as taxonomic lumping increases with respect to phylogenetic divergence times, the number of hotspot cells successively decline for both birds (**a**) and amphibians (**b**).

In terms of phylogenetic endemism (new **Fig. 3**), we have improved the labels to “global extent”, “continental extent” and “national extent”, as suggested by the reviewer. In addition, we have included simplified visual illustrations of how extent is calculated (see **Fig. 3** below). We have also added this discussion in the main text (see Lines 85–88, 170 onwards).

a Birds

b Amphibians

Fig. 2 | Scale-dependent effects of spatial grain and taxonomic treatment on hotspots of weighted endemism, calculated by weighting species according to their range sizes. a birds ($n = 10,018$ species), and **b** amphibians ($n = 5872$ species). Hotspots were identified using the 2.5% threshold based on the 97.5th percentile values for weighted endemism per grid cell (indicated in red), and 5% corresponding to 95th percentile values (indicated in yellow). Variations of taxonomic treatments in presented results are based on species divergence times at varying time depths: present-day, from 2 (Ma, and from 4 Ma – replicating increasing 'lumping' of taxa. Scatterplots indicate that as taxonomic lumping increases with respect to phylogenetic divergence times, the number of hotspot cells successively decline for both birds (**a**) and amphibians (**b**). Analysis of clade collapse was based on a randomly selected subset of 100 trees from a posterior distribution of 600 trees for birds and 100 trees from a posterior distribution of 10,000 trees for amphibians. The maps are in Behrmann projection. See Supplementary Figure 2 for the full variation of weighted

endemism across grid cells at 50, 100, 200, 400, and 800 km and at varying taxonomic treatment based on species divergence times from 1, 2, 3, 4 and 5 Ma.

a Birds

b Amphibians

Fig. 3 | Hotspots of phylogenetic endemism (geographic concentrations of phylogenetically and geographically restricted species) are influenced strongly by spatial extent, varying along global, continental and local extents at country level. a birds ($n = 10,018$ species), and **b** amphibians ($n = 5872$ species) of the world across three levels of spatial extent (national, continental, and global). Global patterns of phylogenetic endemism reflect diversification processes but are lost at the regional to country extents. When extent is varied, we are varying the scale phylogenetic endemism is calculated. Also pictured on the left in the inset are simplified visual illustrations of how extent is calculated. Hotspots were identified using the 2.5% threshold based on the 97.5th percentile values for phylogenetic endemism per grid cell (indicated in red), and 5% corresponding to 95th percentile values (indicated in yellow). The maps are in Behrmann projection. See Supplementary Figure 3 for full variation of phylogenetic endemism across grid cells at 50, 100, 200, 400, and 800 km.

Figure 3. I have several issues here, and I hope that I am misunderstanding something, because if I am not, the results might be a bit different from what the authors state. Please double check. Specifically:

- First, I don't see how the effect is "overwhelming" at fine scales (as described in caption and in the main text). It actually seems closer to zero than at coarse scales, but I may be missing something.

RESPONSE

We apologize that our text was not clear enough. The results reported are correct and the wording has been revised, but the y-axis was log-transformed to normalize the data.

- Second, and forgive me if I'm wrong, but is the y-axis really log endemism? Isn't it some sort of effect size? Guessing from the range of values, it seems like these are standardized model coefficients (but if this is the case, then the effects are mostly quite weak).

RESPONSE

In this revision, we have clarified that the y-axis is a log transformation of endemism values rather than effect size. This transformation is to make the results comparable across scales. The axis label has been revised to improve clarity.

- Third, some points have no confidence intervals. Is this because they are narrower than the point size, or is there an error?

RESPONSE

Yes, these points have narrower confidence intervals suggesting that they are narrower than point size. In addition, the analysis was run across a posterior distribution of trees before taking the median value across cells to account for phylogenetic uncertainty. In addition, we have improved the clarity of the legend by adding supplementary tables (Tables S1 and S2) with more information on the confidence intervals and P values. We have also added a line in the figure legend stating:

"The y-axis was log-transformed to normalize the data. Note that no confidence intervals are visible in the plot since they are narrower than the width of the point size (see Supplementary Tables S1 and S2)."

- Fourth, you might want to add a gray or dashed horizontal line cutting through zero (if y-axis is effect size).

RESPONSE

Amended (see revised figure below, now **Fig. 4**, including a horizontal dotted line cutting through zero).

Fig. 4 | Changes in patterns of weighted endemism and phylogenetic endemism in relation to heterogeneity in environmental variables at different spatial grain. a birds and b amphibians. Weighted endemism was calculated by weighting species according to their range sizes, whereas phylogenetic endemism weights phylogenetic branches by the inverse of the geographical range size of that branch, such that the branch lengths of range-restricted species are given more weight. Statistical analysis is based on a mixed effects model using spatial autocovariate regression models using standard deviation of environmental heterogeneity for each grid cell. These models indicate that our findings are explained by environmental heterogeneity at finer grains (temperature, precipitation, elevation, productivity), and to a far lesser extent at coarser resolutions. Error bars represent lower and upper confidence intervals. The standard errors at the 800 km x 800 km resolution go outside the scale. The y-axis was log-transformed to normalize the data. Note that no confidence intervals are visible in the plot since they are narrower than the width of the point size (see Supplementary Tables 1 and 2).

MINOR COMMENTS

Line 30: I don't think that diversification processes belong to the abstract. They are not what you directly study.

RESPONSE

Removed. Wording has now been revised:

“Global patterns of PE can provide insights into complex evolutionary processes, but this congruence is lost at the regional to country extents.”

Figures 3, 4, 5 and maybe other figures (and tables) - I would add what "weighted" means, i.e. what it is weighted by. This is because figures and their captions should be self-explanatory,

understandable without the need to the main text. Many will jump directly to the figures, skipping the main text.

RESPONSE

Accordingly, we have revised the wording of the figure legends, including at definition of “weighted” (underlined here for emphasis). Some of the revised captions are shown here:

“Fig. 2 | Scale-dependent effects of spatial grain and taxonomic treatment on hotspots of weighted endemism, calculated by weighting species according to their range sizes...”

“Fig. 3 | Hotspots of phylogenetic endemism (geographic concentrations of phylogenetically and geographically restricted species)...”

“Fig. 4 | Changes in patterns of weighted endemism and phylogenetic endemism in relation to heterogeneity in environmental variables at different spatial grain. a birds and b amphibians. Weighted endemism was calculated by weighting species according to their range sizes, whereas phylogenetic endemism weights phylogenetic branches by the inverse of geographical range size of that branch, such that the branch lengths of range-restricted species are given more weight. Statistical analysis is based on a mixed effects model using spatial autocovariate regression models using standard deviation of environmental heterogeneity for each grid cell. These models indicate that our findings are explained by environmental heterogeneity at finer grains (temperature, precipitation, elevation, productivity), and to a far lesser extent at coarser resolutions. Error bars represent lower and upper confidence intervals. The standard errors at the 800 km x 800 km resolution go outside the scale. The y-axis was log-transformed to normalize the data. Note that no confidence intervals are visible in the plot since they are narrower than the width of the point size (see *Supplementary Tables S1 and S2*).”

“Fig. 5 | Relationship between variation in grain size and the proportion of endemism hotspots covered by the global systems of Protected Areas. a bird weighted endemism (left) and phylogenetic endemism (right), and b amphibian weighted endemism (left) and phylogenetic endemism (right). Weighted endemism was calculated by weighting species according to their range sizes, whereas phylogenetic endemism weights phylogenetic branches by the inverse of geographical range size of that branch, such that the branch lengths of range-restricted species are given more weight. The dotted lines represent the 10% threshold corresponding to the minimum representation target for sustaining species persistence. These findings demonstrate widespread deficits of protection for endemism hotspots regardless of grain size, spatial extent or taxonomic treatment. The bottom and top of boxes show the first and third quartiles respectively, the median is indicated by the horizontal line, the range of the data by the whiskers.”

While reading your paper, I did a little search for literature, just to see what is out there, and I skimmed through some of my papers and the citations within. The main conclusion is that your contribution does indeed seem new. However, I thought that maybe it could be useful/inspirational to give you some other references that seem relevant - here it's completely up to you whether to include them or not (by all means avoid “including citations to please the reviewer” situation).

- Ochoa-Ochoa 2014 Biol. J. Linn. S., 10.1111/bij.12201 deal with spatial patterns of endemism, and their scale-dependency.
- IUCN red list guidelines, <https://cmsdocs.s3.amazonaws.com/RedListGuidelines.pdf>, Figure 4.6 and related sections, and citations therein. I think that this is the most practical and applied case where the scale-dependency of range size is used. Since range size is the key ingredient in determining endemism, it seems highly relevant. For a more comprehensive treatment of the grain-dependency of range size and extinction risk see the many works by William Kunin.
- Townsend-Peterson & Watson 1998 Div. Distr. 10.1046/j.1472-4642.1998.00021.x. This seems like something that you might want to check/cite, or see who cites it.
- Keil et al. 2015 Nature Communications, 10.1038/ncomms9837. Their Figure 3 shows how endemism and phylogenetic endemism scales with area (i.e. the endemic-area curve, EAR, and phylogenetic endemism area curve, PDXAR), and how taxonomic and phylogenetic endemism are related. Their Figure 4 shows how this scaling varies across continents.
- Graham et al. 2018 GEB 10.1111/geb.12686 seems like something that you should definitely look at and cite it, since it is tightly related to your “taxonomic lumping”.
- Storch & Sizzling 2008 Folia Geobotanica 10.1007/s12224-008-9015-8 deal with similar issues.
- I am not Storch.

RESPONSE

Thank you for these suggestions. We have examined these references and agreed that they all contributed to our discussion, so we have incorporated them into our discussion of general patterns of endemism e.g. see Lines 51–53:

“It has also been suggested that the effects of scale may be common in patterns of endemism^{21–26}, yet there has never been, to the best of our knowledge, a global assessment of this phenomenon.”

Fig. 4 – State what the box and whiskers show.

RESPONSE

The bottom and top of boxes show the first and third quartiles respectively, the median is indicated by the horizontal line, the range of the data by the whiskers. This figure legend has now been revised as **Fig. 5**:

“Fig. 5 | Relationship between variation in grain size and the proportion of endemism hotspots covered by the global systems of Protected Areas. a bird weighted endemism (left) and phylogenetic endemism (right), and b amphibian weighted endemism (left) and phylogenetic endemism (right). Weighted endemism was calculated by weighting species according to their range sizes, whereas phylogenetic endemism weights phylogenetic branches by the inverse of geographical range size of that branch, such that the branch lengths of range-restricted species are given more weight. The dotted lines represent the 10% threshold corresponding to the minimum representation target for sustaining species persistence. These findings demonstrate widespread deficits of protection for endemism hotspots regardless of grain size, spatial extent or taxonomic treatment. The bottom and top of boxes show the first and third quartiles respectively, the median is indicated by the horizontal line, the range of the data by the whiskers.”

Fig. M1 – Replace the “ANALYSIS” label in the figure by either extent, or grain, whatever is appropriate. Also, try to think about how to migrate a version of such figure to the main text that would illustrate all your important concepts and scalings.

RESPONSE

Revised. We have migrated this into **Fig. 1** in the main manuscript.

Lines 147 – 153: The sentence is too long and complicated.

RESPONSE

Reworded

Lines 203 – 206. Please state that you used standard deviation of these variables, not their mean values.

RESPONSE

Revised:

“To test whether environmental heterogeneity influences patterns of weighted and phylogenetic endemism, we used standard deviation of four commonly used environmental predictors: elevation, annual temperature (temperature henceforth), annual precipitation (precipitation henceforth) and net primary productivity (productivity henceforth).”

I really appreciate that the code and data are publicly available. Great job!

RESPONSE

Thank you for the positive remark.

Reviewer #2 (Remarks to the Author):

This MS covers an interesting issue, of quite some interest for how to use biodiversity data to prioritise conservation actions. It analyses large phylogenies in combination with large online distribution databases. The question of scale-dependence and species level taxonomic has been covered in several other papers, but this MS represents an important extension, since it is done for the whole world, and it presents an objective measure of degree of systematic ‘lumping’.

RESPONSE

We appreciate the reviewer’s positive remarks.

Here, Fig. S2 is very illustrative.

However, the MS also contains some very strange results, which require better explanation and interpretation. The Discussion seems to suffer from an insufficient understanding of the global variation in species diversity.

RESPONSE

This is an important point. We have improved our discussion of the results in the context of spatial variation in diversity (see Lines 89–103 of the Introduction and Lines 180–183 of the Discussion).

The maps of phylogenetic endemism (Fig. 2 and S3) initially did not make sense to me, especially the category ‘national’, which shows red grid-cells in areas that only have few and widespread species (especially in the Western Palearctic, north of the Alps; note that Europe has only two Amphibia species with restricted distributions outside the Mediterranean Area). When I came to the Data Analysis section I finally began to understand what is the problem, and I need to go a bit into detail here to explain my interpretation of this seemingly odd result: Some of these grid-cells are in areas with only two or three widespread amphibian species, so this is a very small sample, and I think that these areas were recognized as ‘areas of phylogenetic endemism’ because they combine wide species distributions and phylogenetically highly overdispersed communities. Birds are less extreme, but European species (outside the Mediterranean area) are generally widespread, but again, the

communities are generally phylogenetically overdispersed, leading to some grid-cells being detected as areas of high 'phylogenetic endemism'. Note that Europe N of the Alps is generally characterized by widespread and phylogenetically independent species re-colonizing the area over and over again (during interglacial periods) from Pleistocene refuges in the South and in Asia (the only exception is *Loxia scotica*, of doubtful species rank) (quite unlike in the Eastern Palearctic, which mainly remained ice-free during the Pleistocene, and therefore was an area of continued speciation. This general pattern, with predominance of old species in the Western Palearctic (and in some tropical and austral parts of the world), is illustrated for Passerine birds in Chinese Birds 4 (2013): 132-43.

RESPONSE

Thank you for these comments, they are appreciated. The 'national' results for both birds and amphibians were calculated by mapping global phylogenetic endemism before taking the hotspots within the boundaries of countries and not per continent. This was done to illustrate that when PE is calculated at too narrow a scale, odd results may appear as the reviewer is highlighting for the Western Palearctic. However, insights into more complex evolutionary processes such as dispersal, speciation and extinction shaping biodiversity patterns are captured at larger continental to global extents (see Lines 180–183). We have clarified this in the Methods in Lines 699–702.

In addition, we now provided a conceptual figure as a new **Fig. 1** (we have dealt with this above in response to the similar comment from Reviewer 1) plus additional schematic cartoons to accompany **Figs. 2 and 3** to make what we hope is a much clearer conceptual illustration of our predictions of empirical patterns of endemism at varying scales.

In order to avoid confusion over how these strange geographical patterns appear, I think, first of all, that the Introduction should provide a more transparent explanation of what the 'phylogenetic endemism' means (line 54). You could simply move the explanation on lines 520-522 to the Introduction. Maybe even with an additional statement, that this means that even areas with quite widespread species are detected as areas of phylogenetic endemism if the local communities are phylogenetically highly overdispersed. You could also remove the problem with too small samples by removing areas with very few (for instance <10) species. The 'country level' aspect I find quite odd. Most countries like to flag, in their national biodiversity reviews, their list of endemic species (recorded in no other nation), and it could be interesting to see to what extent this would be affected by taxonomy. But I fail to see the relevance of analyzing the occurrence of small marginal populations within a national territory of species that are widespread, common and in the 'lower risk category' outside that national territory. So what is the relevance of emphasizing (line 278) of the tiny population of ringed kingfisher in Texas, when this species is distributed almost all over Central and South America? IUCN has had great problems with agreements over criteria for national red-lists for species that are not globally threatened, and I don't think that the inclusion of national concerns over marginal populations adds much to an otherwise interesting analysis.

RESPONSE

We feel that this is a particularly insightful observation. In this manuscript, we have improved the definition of phylogenetic endemism by moving the explanation of phylogenetic endemism from the Methods to the Introduction (Lines 56–60), including the discussion on how widespread species can influence patterns of phylogenetic endemism if local communities are phylogenetically overdispersed.

However, the removal of samples of too small a size, such as <10 species, would be too arbitrary, and would probably raise many additional questions beyond the key message of the study. Regarding the oddity of the 'country level' analysis, we draw your attention to our in-depth

response to a similar comment above at length (page 13 of this document), where we indicated that the national results were calculated by mapping global phylogenetic endemism before taking the hotspots within the boundaries of countries and not per continent. This is to demonstrate that when species ranges span socio-political borders, such that PE is calculated regionally (within a continent or country) without consideration of a species' full range, an inflation of phylogenetic endemism results. We have added some discussion of this in the figure legend of **Fig. 1**.

Lines 203-226: I don't think there is much novel in this analysis. It is well established that landscape heterogeneity is an important factor explaining differentiation of local populations and thereby the degree of endemism. A particularly strong demonstration of this is by Rahbek et al. in *Science* 365 (2019): 1108-1113. But it is important to note that the pattern is erased in regions with highly seasonal climates, which explains why high endemism is so strongly concentrated to the tropics, and mainly to areas under the climatic influence of thermally stable tropical oceans. If the authors could demonstrate how correlation between endemism and landscape heterogeneity depends on latitude, that would be a novel element.

RESPONSE

This is a great point. We agree strongly that it is of value to explore the correlation between endemism and landscape heterogeneity and how it depends on latitude. We believe this is a major contribution of this revised manuscript. We pursue that issue in some detail by running new analysis of the correlation between endemism and landscape heterogeneity (measured as standard deviation of altitude per grid cell) versus latitude. Results show that residuals of landscape heterogeneity and endemism indeed concentrate in the tropics and decline with latitude (see figure below).

As noted in the figure below (new Figure S4), we found that the residuals of endemism vs. landscape heterogeneity are highest in the tropics and decline with latitude. This trend is particularly strong at coarser grain size. We have added this figure in the supplementary material and a discussion of it in the Discussion in Lines 242–248.

“When we compared the relationship between endemism and landscape heterogeneity (measured as heterogeneity of altitude above sea level) versus latitude, we found that the residuals of landscape heterogeneity and endemism concentrate in the tropics and decline with latitude (Figure S4). This trend is particularly strong at coarser grain size. Patterns of endemism might therefore have been erased in regions with highly seasonal climates, corroborating the high endemism in the tropics, and mainly in areas under the climatic influence of thermally stable tropical oceans⁷⁴.”

Figure S4 | Scatterplot of latitudinal variation with residuals from landscape heterogeneity and phylogenetic endemism. Landscape heterogeneity was measured as standard deviation of altitude per grid cell. Trend line (in red)

computed by evaluating the loess smooth at equally spaced points covering the range of endemism values for each latitude.

Other relevant papers would be by Rahbek et al. in Proc. Royal Soc. B 274(2007): 165-175 regarding the analysis of scale-dependence, and Dillon & Fjeldså in Ecography 28 (2005): 682-692 regarding significance of how species are defined.

RESPONSE

We thank the reviewer for these additional articles, which we have included in this revision in Line 50 and Line 71.

Details: In line 25, I don't think the problem is 'poorly known', but it has not been analyzed in a systematic way. So maybe use the word 'analyzed' instead of 'known'.

RESPONSE

We have made this correction.

Reviewer #3 (Remarks to the Author):

The paper by Daru and collaborators investigate the relationship between endemism patterns (Weighted Endemism and Phylogenetic Endemism) and spatial scale (i.e. grain size). I think the manuscript is well written and the methods and analyses are sound. This is a stimulating topic and should be of interest to a broad audience of readers of Nature Communications. I only have a few concerns/comments that I think the authors need to clarify. I think these editions would improve the manuscript.

RESPONSE

We thank the reviewer for the positive remarks. We have revised several areas of the introduction and discussion with the aim of improving the clarity.

Line 52-66 – I think these issues are too methodological. The previous paragraph does a good job explaining that all biodiversity patterns are scale-dependent and that different mechanisms may operate on different scales. I think at the beginning of the Introduction the authors should better explore the broader context of what are the potential mechanisms that might explain variation in endemism across scales.

RESPONSE

This is an important point. We have now added a discussion in the introduction on potential mechanisms that can drive patterns of endemism across scales, in Lines 89–103:

“Areas of endemism represent important units for postulating hypotheses in historical biogeography⁴⁴⁻⁴⁶, and are priority targets for conservation action because they capture facets of biodiversity not represented elsewhere^{31,32,47,48}. For example, areas that have experienced higher historical temperature change tend to harbour fewer endemic species, often with phylogenetically derived species (neo-endemics) occupying higher latitudes^{49,50}. In contrast, climatic shifts that lead to low levels of change in species' geographical distributions may allow the survival of ancient lineages that have become extinct elsewhere (paleoendemics)⁵¹. Therefore, we predict that the local extinction of a paleoendemic lineage can increase patterns of phylogenetic endemism, whereas the loss of a neoendemic will have less impact on phylogenetic endemism, at least initially. Only by losing entire clades will the loss of neoendemics result in a significant change in phylogenetic diversity. A high dispersal rate will cause fewer species to be confined to a

specific area, leading to lower concentration of endemic species⁴⁹. Conversely, the phylogenetic composition of communities including species with poor dispersal abilities will cause the aggregation of close relatives, leading to increased phylogenetic endemism⁵².”

Line 67-69 – It is not clear why PE “offers a potential solution to deal with new taxonomic knowledge in conservation strategies because some lineages and areas harbor far more endemic diversity than would be expected from species ranges alone”

RESPONSE

This section has been revised to improve clarity:

“In contrast, phylogenetic endemism offers a potential solution to deal with new taxonomic knowledge in conservation strategies. This is because phylogenetic endemism is not greatly influenced by oversplitting of neoendemics (more phylogenetically derived species), for example, if populations only isolated since the last ice age are elevated to species level.”

Line 69-82 – I also think that this paragraph should bring a more general account of what processes could lead to the patterns seen in PE at different scales

RESPONSE

We agree with this, and we discuss how we have dealt with this above in response to the Reviewer's similar comment (see Lines 89–103).

Line 88-92 – These predictions are not very clear to me. In the amphibian example, you mentioned that the effect of scale on endemism should be strongest at finer grains. Isn't the scale effect necessarily a change across scales?

RESPONSE

We have revised this sentence to reflect that amphibian endemism changes across spatial grain and extent:

“On the other hand, amphibians are poor dispersers and possess reduced geographic ranges compared to birds⁵⁷, and thus we predict the effect of scale on endemism in amphibians to change across spatial grain and extent.”

Line 93-95 – I would argue that this is not a very interesting way of framing your main hypothesis and therefore the main goal of your article. As you stated in the first paragraph all biodiversity patterns are influenced by spatial scale.

RESPONSE

Revised to improve clarity:

“Here, we use comprehensive datasets on the phylogenetic relationships and geographic distributions for c. 10,000 species of birds and 6,000 species of amphibians across the globe to test the hypothesis that changes in taxonomic treatment, spatial grain and extent can influence patterns of weighted endemism and phylogenetic endemism.”

Line 100-104 – I think these are interesting questions but the previous section of the introduction did not properly provide the necessary contextualization related to these questions (except for some text regarding question i). It would be better to have a better contextualization of these topics and some predictions.

RESPONSE

This is an important point. We have provided some background on potential mechanisms that could underlie patterns of endemism in Lines 89–103 to contextualize these topics.

Line 113-115 – Perhaps ‘richest’ is not the best word here since it could confuse the reader. From the figure, it is difficult to see what the authors refer to. Perhaps a table containing the number of grid cells in the high 2.5% could help. Also, as the cell size increases the number of total cells decreases so, the decrease in the number of cells with highest 2.5% WE could be proportional to this total number of cells decrease?

RESPONSE

We completely agree and have revised this section to improve clarity. Please see our response above to Reviewer 1's point, where we address this comment in depth:

“As species are treated by taxonomic lumping based on their divergence times at varying time depths, our results show that the number of hotspot cells (grid cells with the 97.5th percentile values for weighted endemism) successively decline with increasing spatial grain (Fig. 1), because species lumping collapses smaller ranges into fewer larger parts.”

Part of our response is the addition of a new figure (Fig. 2) that incorporates scatterplots along with R-squared values showing the number of hotspot cells against taxonomic lumping in terms of divergence times.

Line 121-123 – Is this what the data shows? Did you observe a loss of hotspots of endemism in islands and mountain tops?

RESPONSE

Because we have rewritten large parts of the intro and discussion, this sentence has been rewritten in the revised manuscript to improve clarity.

Line 127-131 – This answered my question above so I think you should combine this somehow.

RESPONSE

We separated this paragraph from the previous paragraph to provide a nice break between the findings, thereby avoiding overly long paragraphs. However, we are happy to combine the paragraphs if the editor deems necessary.

Line 139-141 – Is there anything more you can say about the taxonomic issue? It seems trivial that using different concepts would produce different results.

RESPONSE

This section has been revised to improve clarity:

“While consistency in species concepts has been advocated in macro-scale studies (e.g. ref.³⁹), our results show that using the biological or phylogenetic species concepts can produce different results and might influence conservation prioritization differently.”

Line 148 – I suggest replacing the term ‘biodiversity hotspot’ here. This can confuse the reader since is a well-know term in other literature.

RESPONSE

We have changed the wording here.

Line 155-158 – This sentence is too generic it would be more helpful to state how PE can influence on those things.

RESPONSE

We point to the Reviewer's previous comment about potential mechanisms that can underlie patterns of endemism:

“Areas of endemism represent important units for postulating hypotheses in historical biogeography^{44–46}, and are priority targets for conservation action because they capture facets of biodiversity not represented elsewhere^{31,32,47,48}. For example, areas that have experienced higher historical temperature change, tend to harbour fewer endemic species, often with phylogenetically derived species (neo-endemics) occupying higher latitudes^{49,50}. In contrast, climatic shifts that lead to low levels of change in species’ geographical distributions may allow the survival of ancient lineages that have become extinct elsewhere (paleoendemics)⁵¹. Therefore, we predict that the local extinction of a paleoendemic lineage can increase patterns of phylogenetic endemism, whereas the loss of a neoendemic will have less impact on phylogenetic endemism, at least initially. Only by losing entire clades will the loss of neoendemics result in a significant change in phylogenetic diversity. A high dispersal rate will cause fewer species to be confined to a specific area leading to lower concentration of endemic species⁴⁹. Conversely, the phylogenetic composition of communities including species with poor dispersal abilities will cause the aggregation of close relatives leading to increased phylogenetic endemism⁵².”

Line 203-206 – Earlier in the text the authors need to provide a succinct account on how these environmental variables are expected to influence endemism and how they should change with scale.

RESPONSE

In response to a similar comment above on mechanisms that might underlie patterns of endemism, we have added a statement on how environmental conditions can influence patterns of endemism:

“...climatic shifts that lead to low levels of change in species’ geographical distributions may allow the survival of ancient lineages that have become extinct elsewhere (paleoendemics)⁵¹.”

Reviewers' Comments:

Reviewer #1:

Remarks to the Author:

I liked the first submission and I like the revision even better. The authors have well addressed all of my concerns, and I now have only minor comments. Otherwise I think that the paper is good to go.

MINOR COMMENTS

Lines 149-150: Something about the sentence does not feel right.

I really like Figure 1. The only thing that is not clear is what exactly is Site 1 and Site 2. Is it the center of the X symbol, or its boundary?

I still don't understand Figure 4. If the y-axes really are endemism (and not effect sizes, as was my concern in the first version), then how are they related to the environmental variables? Also, I asked the authors to add a horizontal line to the figure, but that was because I believed that the plots show effect sizes (i.e. model coefficients); now the authors claim that these are not effect sizes, but actual log transformed values of endemism. If this is the case, then why have they still provided the horizontal lines? And why does the y axis have a span between -1 and 1 – that would indicate some sort of standardized coefficients. Finally, where do the confidence intervals come from – usually confidence intervals are reserved for parameter estimates, i.e. model coefficients. If the y-axes really are endemism values, then they are maybe some sort of partial residual effects perhaps? Hard to tell, this is really confusing. I think that something either needs to be explained, or fixed here. In other words, it needs to be clear how the four colourful lines in each panel were produced – it is not very informative to state that "Statistical analysis is based on a mixed effect model". I am sorry, but now it is not clear. Or I am really missing something, but then prove me wrong. Also, if this is fixed, then I believe that the figure caption can be much shorter – if you clearly state what y-axes are, then there will be little need for complex descriptions. And finally, maybe state that the 4 environmental variables are actually standard deviations directly in the figure.

Otherwise I congratulate the authors, and I am looking forward to seeing this published.

Reviewer #2:

Remarks to the Author:

The MS has now been much improved, first of all by providing a better explanation already in the Introduction (with a conceptual figure) of various methodological issues and what is meant by scale-dependent and weighted endemism. Also, the terminology (and use of the terms 'scale' and 'grain') is now clear. In general, I am satisfied with how the authors responded to the earlier comments from reviewers.

Tiny edits: In line 250: change 'increase' to 'increased' or 'an increase in'.

REVIEWERS' COMMENTS:

Reviewer #1 (Remarks to the Author):

I liked the first submission and I like the revision even better. The authors have well addressed all of my concerns, and I now have only minor comments. Otherwise I think that the paper is good to go.

RESPONSE

We thank the reviewer for the positive remarks. In this final version, we have revised the manuscript taking in consideration all of the reviewer's remaining comments.

MINOR COMMENTS

Lines 149-150: Something about the sentence does not feel right.

RESPONSE

We appreciate the reviewer's comments. In revising the manuscript to more clearly describe the pieces that were novel to the present analysis, we have removed this line without losing meaning to the original findings of the paragraph.

I really like Figure 1. The only thing that is not clear is what exactly is Site 1 and Site 2. Is it the center of the X symbol, or its boundary?

RESPONSE

In this revision, we have improved clarity in this figure by substituting the X with a circle. Therefore, sites 1 and 2 refer to the area of the circle overlapping with the cells (species ranges) underneath. This is now more clearly explained in the figure caption (see figure below; see also underlined text in figure caption below for emphasis).

Weighted endemism

Phylogenetic endemism

Fig. 1 | Schematic of the variation of endemism with spatial extent, grain size and taxonomic treatment. **a** and **b**, Changes in weighted endemism (species richness inversely weighted by species ranges) under a scenario of taxonomic lumping at a divergence time of 1 million years ago (Ma). As the node and ranges that originated at 1 Ma were collapsed (**b**), new hotspot maps of weighted endemism were generated which we compared to the original data (**a**). **c** Variation of phylogenetic endemism (the degree to which phylogenetic diversity is restricted to any given area) with spatial extent. **d** Spatial distribution of phylogenetic endemism across a global extent. At a global extent, PE is calculated accounting for the full geographic range of the species. **e** Distribution of phylogenetic endemism (PE) at a regional extent (continent or country). When species ranges span socio-political borders such that PE is calculated regionally (within a continent or country) without consideration of a species' (or even a clade's) full range,

an inflation of phylogenetic endemism results. Sites 1 and 2 refer to the area of circle overlapping with the cells (species ranges) underneath. Source data are provided as a Source Data file.

I still don't understand Figure 4. If the y-axes really are endemism (and not effect sizes, as was my concern in the first version), then how are they related to the environmental variables? Also, I asked the authors to add a horizontal line to the figure, but that was because I believed that the plots show effect sizes (i.e. model coefficients); now the authors claim that these are not effect sizes, but actual log transformed values of endemism. If this is the case, then why have they still provided the horizontal lines? And why does the y axis have a span between -1 and 1 – that would indicate some sort of standardized coefficients. Finally, where do the confidence intervals come from – usually confidence intervals are reserved for parameter estimates, i.e. model coefficients. If the y-axes really are endemism values, then they are maybe some sort of partial residual effects perhaps? Hard to tell, this is really confusing. I think that something either needs to be

explained, or fixed here. In other words, it needs to be clear how the four colourful lines in each panel were produced – it is not very informative to state that “Statistical analysis is based on a mixed effect model”. I am sorry, but now it is not clear. Or I am really missing something, but then prove me wrong. Also, if this is fixed, then I believe that the figure caption can be much shorter – if you clearly state what y-axes are, then there will be little need for complex descriptions. And finally, maybe state that the 4 environmental variables are actually standard deviations directly in the figure.

RESPONSE

Many thanks for your insightful and detailed comments on this figure. The four environmental variables are standard deviations of each variable within hotspots cells. For each hotspot cell and grain size, we extracted the standard deviation of four environmental variables (temperature, precipitation, elevation, productivity). Because the standard deviations for the four environmental variables are in different units, we log transformed them for comparison. We then ran a mixed effects model correcting for spatial autocorrelation using log transformed endemism values within the hotspots cells as response variable (y-axis) and the standard deviation of environmental variables per cell as explanatory variables. The fixed effects were then used to compare the likelihoods of the fitted objects. The final figure was generated by plotting estimated values for each environmental factor and grain size. We have also provided the source data underlying this figure as a Source Data file in the tabs Fig4a–4b. The key points in this explanation has been added to the caption (see below).

Fig. 4 | Changes in patterns of weighted endemism and phylogenetic endemism in relation to heterogeneity in environmental variables at different spatial grains. **a** birds ($n = 10,018$ species) and **b** amphibians ($n = 5872$ species). Weighted endemism was calculated by weighting species according to their range sizes, whereas phylogenetic endemism weights phylogenetic branches by the inverse of geographical range size of that branch, such that the branch lengths of range-restricted species are weighted higher. For each hotspot cell and grain size, we extracted the standard deviation of four environmental variables (temperature, precipitation, elevation, productivity) and ran a mixed effects model correcting for spatial autocorrelation using log transformed endemism values within the hotspots cells as response variable (y-axis) and the standard deviation of environmental variables per cell as explanatory variables. The fixed effects were then used to compare the likelihoods of the fitted objects. These models indicate that our findings are explained by environmental heterogeneity at finer grains, and to a far lesser extent at coarser resolutions. Error bars represent lower and upper confidence intervals ($n = 20$ for the four environmental variables across grain sizes). The standard errors at the $800 \text{ km} \times 800 \text{ km}$ resolution go outside the scale. Note that no confidence intervals are visible in the plot since they are narrower than the width of the point size (see Supplementary Tables 1 and 2).

Otherwise I congratulate the authors, and I am looking forward to seeing this published.

RESPONSE

We thank the reviewer for the positive remarks.

Reviewer #2 (Remarks to the Author):

The MS has now been much improved, first of all by providing a better explanation already in the Introduction (with a conceptual figure) of various methodological issues and what is meant

by scale-dependent and weighted endemism. Also, the terminology (and use of the terms 'scale' and 'grain') is now clear. In general, I am satisfied with how the authors responded to the earlier comments from reviewers.

Tiny edits: In line 250: change 'increase' to 'increased' or 'an increase in'.

RESPONSE

We appreciate the reviewer's positive remark and have we have followed the reviewer's suggestion.

Reviewers' Comments:

Reviewer #1:

Remarks to the Author:

MAJOR COMMENTS

Dear authors, really like the manuscript, but the mixed effects part and the associated Figure 4 really need to be fixed before it is accepted. I think that it is also in your interest to get this part right, because it will save you from potential criticism after publication. Specifically:

1. The methods related to mixed effect modelling and calculation of endemism described in Fig. 4 should go to the Methods section and they should be properly describe there. Also, moving the mixed-effect model description to their own methods paragraph would make the already bloated figure caption much more readable. I think that it should take you not more than 2 paragraphs to describe the modelling correctly. More specifically:

- Have the general mathematical formula of the models, or the R code line, in the methods. This is usually the most effective way to clarify any statistics-related confusion. In the formula, you can then identify the variables and/or coefficients that are described in the figure.
- State how the spatial component was modeled exactly. Was it modelled in the mean response, in the residuals? Did you use an autoregressive model or a smooth spline or something else?
- How was the model fitted (maximum likelihood, MCMC, ...)? Which package/software?
- Do you use all of the predictors in one model, or do you show results for four independent models? If the former is true, can you provide some information criteria to compare the models (e.g. AIC, BIC, DIC, WAIC, ...)? Also, can you give the overall fractions of explained variation?
- If the environmental variables used in the models are actually their standard deviations, then don't refer to these variables as simply "temperature". Rather, use "temperature SD", or "temperature variation", or something that would make it clear right in the actual figure
- Clarify what was the random and the fixed part.

2. I did read your response and the updated caption several times. And a note here: I do teach a course on hierarchical and mixed-effect models, and I have published several papers that use them. Yet I still have no idea what the y-axes in Figure 4 are, and how you've got the numbers. There are statements like "fixed effects were then used to compare likelihoods of the fitted objects", or "the figure was generated by plotting estimated values for each environmental factor" which are either statistically wrong, or unclear. Note that:

- It is not clear how fixed effects can be used to compare likelihoods. Likelihood is joint probability density of observing the data, given the entire model. Please fix the language (plus, this statement definitely does not belong to the figure caption).
- It is unclear what "estimated values for each environmental factor are", mostly because it is unclear if these are from separate models, or if these are marginal effects from a single model with all predictors in it. Also, just specify it better, i.e. "average estimated values of endemism predicted by a model with the given environmental predictor" (but I have no idea if this is what you did, because I don't understand your methods). Just state it exactly.
- Confidence intervals of what? Estimated mean? Or are these actual prediction intervals (which would appear so given that they seem to be based on predictions)? Note that a prediction interval is something fundamentally different from confidence interval of the mean, and these are something very different from standard errors, which brings me to your statement that "standard errors at the 800 km x 800 km resolution ... confidence intervals are visible" - so are these error bars or confidence intervals? And again, what do they relate to? The mean?
- Given the scale of the y axis (-1 to 1), and the similarity of the values across mammals and amphibians, I still suspect that you may be plotting some sort of model coefficients, or partial (marginal) effects. I am sorry, but given the overall mess of this whole part or the ms, please convince me better that y-axis is really what you claim it to be.
- If y-axis really represents predictions, then it should be reflected in the y-axis labels. But then why

would you summarize it using a mean and the bars?

3. Perhaps most importantly of all: You state in the caption: "These models indicate that our findings are explained by environmental heterogeneity at finer grains, and to a far lesser extent at coarser resolutions." I fail to see how to see this is actually the figures. If the y-axis are the predictions, then it makes little sense. If y-axis are effect sizes (some sort of model coefficients), then it also seems weird. What I see is that at fine grains your colourful lines (whatever they mean I have no idea) are close to 0, and they become much more variable at coarse grains. But how does this relate to your statement?

I apologize if my criticism sounds harsh, but I just can't get rid of the feeling that this part of the ms is rushed and confused. At the same time, this is actually a routine modelling exercise, and for an average statistician this would be an hour of work, including writing the methods properly. Please take the time, or ask a statistician for help, and get this really right.

Reviewer #1 (Remarks to the Author):

MAJOR COMMENTS

Dear authors, really like the manuscript, but the mixed effects part and the associated Figure 4 really need to be fixed before it is accepted. I think that it is also in your interest to get this part right, because it will save you from potential criticism after publication. Specifically:

1. The methods related to mixed effect modelling and calculation of endemism described in Fig. 4 should go to the Methods section and they should be properly describe there. Also, moving the mixed-effect model description to their own methods paragraph would make the already bloated figure caption much more readable. I think that it should take you not more than 2 paragraphs to describe the modelling correctly. More specifically:

- Have the general mathematical formula of the models, or the R code line, in the methods. This is usually the most effective way to clarify any statistics-related confusion. In the formula, you can then identify the variables and/or coefficients that are described in the figure.

RESPONSE

Many thanks for your positive feedback. We are happy to provide more details as you suggest.

We have moved the description of methodology linked to the mixed-effects modeling to a separate section in the Methods to make Figure 4 more readable. In addition, we have also provided a mathematical formula for the model in the Methods and an R vignette on our new package's website with step-by-step illustration of the analysis (<https://darunabas.github.io/phyloregion/articles/heterogeneity.html>). The section in the Methods describing environmental heterogeneity and mixed-effects model has been revised as follows (Lines 389-441):

“Environmental data and heterogeneity

We selected key environmental factors that are commonly used to examine biodiversity-environment associations. These variables included mean annual temperature, mean annual precipitation, annual net primary productivity and elevation. Mean annual temperature, mean annual precipitation and elevation were downloaded as raster layers from the WorldClim database⁸⁶ at a resolution of 2.5'. Annual net primary productivity was downloaded from NASA Moderate Resolution Imaging Spectroradiometer (MODIS) at a resolution of 1 km and calculated using the MOD17 algorithm. These variables were converted to Behrmann equal-area projection using the function projectRaster in the R package raster⁸⁷.

We defined ‘environmental heterogeneity’ as the variation of environmental factors in each cell, obtained by taking the standard deviation of the environmental variables (temperature, precipitation, elevation and productivity) in each cell and across grain sizes (50, 100, 200, 400 and 800 km). Henceforth, we refer to the standard deviation of these four environmental variables i.e., mean annual temperature, mean annual precipitation, annual net primary productivity and elevation, as temperature variation, precipitation variation, productivity variation and elevation variation, respectively.

Linear mixed-effects model

We fit a single linear mixed-effects model to analyse the effect of environmental heterogeneity on patterns of endemism (PE or WE) across grain sizes. A linear mixed-effects model allows the modeling of data as a combination of fixed effects, random effects and independent random error, and are especially useful when there is non-

independence in the data⁸⁸. The standard form of a linear mixed-effects model is expressed as:

$$Y_i = x_i\beta + s_i + \varepsilon_i$$

where Y_i represents the response variable at grid cell or location i , x_i is a matrix of the observations (explanatory variables) used as predictors (covariates), and β is a vector of the unknown regression coefficients, which are often called fixed effect coefficients, s_i is a matrix, similar to x that captures the complex covariance structure for spatial autocorrelation and ε_i is the random measurement error (residuals). All explanatory variables were standardised prior to statistical analyses so that all variables had a standard deviation of 1 and a mean of 0. This ensures that the estimated coefficients are all on the same scale and for easier comparison of effect sizes.

The linear mixed-effects model was fit as a single model with all the variables in one model predicting endemism (PE or WE) as a function of the four environmental variables (temperature variation, precipitation variation, elevation variation and productivity variation), with continent identity of grid cells as a random effect, allowing us to include any idiosyncratic differences between continents. The model also includes a spatial covariate of geographical coordinates as an additional predictor variable to account for spatial autocorrelation. The spatial covariate was created as a matrix of the coordinates of each cell's centroid corresponding to the geographical Cartesian x/y -coordinates (longitude and latitude), and was calculated with the function `autocov_dist` in the R package `spdep`⁸⁹. For each focal cell, we varied the weighting function and neighbourhood sizes using the next one to two cell neighbours to remove spatial autocorrelation (function `nb2listw` in `spdep`⁸⁹). We used the maximum likelihood optimization criterion over restricted maximum likelihood, to allow significance testing via model comparison.

The linear mixed-effects model was fit in R v.3.6.3⁹⁰ with the `lme` function in the R package `nlme`⁹¹. The variations of environmental variables on endemism are presented as estimated coefficients of the fixed effects and their 95% confidence intervals in the R package `nlme`⁹¹. A vignette, with a worked example, data and R codes describing all the steps for the analyses, is also provided in our R package's website (<https://darunabas.github.io/phyloregion/articles/heterogeneity.html>)."

- State how the spatial component was modeled exactly. Was it modelled in the mean response, in the residuals? Did you use an autoregressive model or a smooth spline or something else?

RESPONSE

This part of the methodology has been revised to improve clarity (we have dealt with this above in our revised description of the Methods). Specifically, the model also includes a spatial covariate of geographical coordinates in the predictors to account for spatial autocorrelation. The spatial covariate was created as a matrix of the coordinates of each cell's centroid corresponding to the geographical Cartesian x/y -coordinates (longitude and latitude), and was calculated with the function `autocov_dist` in the R package `spdep`⁸⁹. For each focal cell, we varied the weighting function and neighbourhood sizes using the next one to two cell neighbours to remove spatial autocorrelation (function `nb2listw` in `spdep`⁸⁹). This information is now clearly provided in the revised manuscript in the Lines 427-433.

- How was the model fitted (maximum likelihood, MCMC, ...)? Which package/software?

RESPONSE

The model was fitted using linear mixed effects model using the function *lme* in the R package *nlme*⁹¹. We used the maximum likelihood optimization criterion over restricted maximum likelihood, to allow significance testing via model comparison. This information is now provided in Lines 433-435 as follows:

“We used the maximum likelihood optimization criterion over restricted maximum likelihood, to allow significance testing via model comparison.”

- Do you use all of the predictors in one model, or do you show results for four independent models? If the former is true, can you provide some information criteria to compare the models (e.g. AIC, BIC, DIC, WAIC, ...)? Also, can you give the overall fractions of explained variation?

RESPONSE

We used all predictors in one model including the autocovariate as additional variable and continent identity for each grid cells as random covariate. In addition, we have included the model output in the Source Data file and a vignette with worked example in our R package’s website (<https://darunabas.github.io/phyloregion/articles/heterogeneity.html>). This information is now provided in Lines 423-433 as follows:

*“The linear mixed-effects model was fit as a single model with all the variables in one model predicting endemism (PE or WE) as a function of the four environmental variables (temperature variation, precipitation variation, elevation variation and productivity variation), with continent identity of grid cells as a random effect, allowing us to include any idiosyncratic differences between continents. The model also includes a spatial covariate of geographical coordinates as an additional predictor variable to account for spatial autocorrelation. The spatial covariate was created as a matrix of the coordinates of each cell’s centroid corresponding to the geographical Cartesian x/y - coordinates (longitude and latitude), and was calculated with the function *autocov_dist* in the R package *spdep*⁸⁹. For each focal cell, we varied the weighting function and neighbourhood sizes using the next one to two cell neighbours to remove spatial autocorrelation (function *nb2listw* in *spdep*⁸⁹).”*

- If the environmental variables used in the models are actually their standard deviations, then don't refer to these variables as simply "temperature". Rather, use "temperature SD", or "temperature variation", or something that would make it clear right in the actual figure

RESPONSE

We have made this correction in the figure and methods (see Lines 401-404):

“Henceforth, we refer to the standard deviation of these four environmental variables i.e., mean annual temperature, mean annual precipitation, annual net primary productivity and elevation, as temperature variation, precipitation variation, productivity variation and elevation variation, respectively.”

- Clarify what was the random and the fixed part.

RESPONSE

The random effects are the continent identities for each grid cell so that the fixed effects coefficients will reflect variation among grid cell observations within a given continent. The fixed effects are the four environmental variables (temperature variation, precipitation variation, elevation variation and productivity variation), with a spatial covariate of geographical

coordinates to account for spatial autocorrelation. This information is now provided in Lines 426-427.

2. I did read your response and the updated caption several times. And a note here: I do teach a course on hierarchical and mixed-effect models, and I have published several papers that use them. Yet I still have no idea what the y-axes in Figure 4 are, and how you've got the numbers. There are statements like "fixed effects were then used to compare likelihoods of the fitted objects", or "the figure was generated by plotting estimated values for each environmental factor" which are either statistically wrong, or unclear. Note that:

- It is not clear how fixed effects can be used to compare likelihoods. Likelihood is joint probability density of observing the data, given the entire model. Please fix the language (plus, this statement definitely does not belong to the figure caption).

RESPONSE

Thanks for this insightful input. We have now made every effort to provide as much clarity as possible on this. We have now clarified that the y-axis corresponds to the **estimates and 95% confidence intervals for the fixed effects** predicted by a linear mixed-effects model of endemism with environmental factors (temperature variation, precipitation variation, productivity variation and elevation variation) across grain sizes. The vertical bars correspond to the 95% confidence intervals for the fixed effects (please see below output from one of the models).

## clim	lower	est.	upper	p_val	scale
## sd_ALT	0.14341347	0.34902185	0.5546302	1.007823e-03	100km
## sd_MAP	-0.01797379	0.04570580	0.1093854	1.620847e-01	100km
## sd_MAT	-0.28232947	-0.08198566	0.1183581	4.249973e-01	100km
## sd_NPP	0.06942198	0.12714706	0.1848721	2.161084e-05	100km

The estimates and 95% confidence intervals for the fixed effects predicted from the model were then plotted for each scale. The source data underlying the figure are included in the Source Data file. This information is now clearly provided in the revised manuscript (in Lines 715-722) and revised Figure (see below):

Fig. 4 | The estimates and 95% confidence intervals for the fixed effects predicted by a single linear mixed-effects model of endemism with all predictors (temperature variation, precipitation variation, productivity variation, elevation variation, spatial autocovariate and random effects of continent identities for each grid cell) across grain sizes. a birds, and b amphibians. These models indicate that our findings are explained by environmental heterogeneity at coarser grains and to a lesser extent at finer resolutions. Significance was assessed by comparing likelihoods of the fitted objects. Source data are provided as a Source Data file.

- It is unclear what "estimated values for each environmental factor are", mostly because it is unclear if these are from separate models, or if these are marginal effects from a single model with all predictors in it. Also, just specify it better, i.e. "average estimated values of endemism predicted by a model with the given environmental predictor" (but I have no idea if this is what you did, because I don't understand your methods). Just state it exactly.

RESPONSE

We apologise that our previous text was not clear enough. In this revision, we have revised wording to reflect that these are the estimates and 95% confidence intervals for the fixed effects predicted by a single linear mixed-effects model of endemism with all predictors (temperature variation, precipitation variation, productivity variation, elevation variation, spatial autocovariate

and random effects of continent identities) in it. The revised figure legend is now clearly provided in the revised manuscript (in Lines 715-722).

- Confidence intervals of what? Estimated mean? Or are these actual prediction intervals (which would appear so given that they seem to be based on predictions)? Note that a prediction interval is something fundamentally different from confidence interval of the mean, and these are something very different from standard errors, which brings me to your statement that "standard errors at the 800 km x 800 km resolution ... confidence intervals are visible" - so are these error bars or confidence intervals? And again, what do they relate to? The mean?

RESPONSE

We have now clarified that these are 95% confidence intervals for the fixed effects predicted from the model, which predicts endemism as a function of all predictors (temperature variation, precipitation variation, productivity variation, elevation variation, spatial autocovariate and random effects of continent identities) across grain sizes. We have clarified this directly in the figure and revised it in the manuscript where it is reflected (see Lines 715-722).

- Given the scale of the y axis (-1 to 1), and the similarity of the values across mammals and amphibians, I still suspect that you may be plotting some sort of model coefficients, or partial (marginal) effects. I am sorry, but given the overall mess of this whole part of the ms, please convince me better that y-axis is really what you claim it to be.

RESPONSE

We again apologize for the lack of clarity in the last submission. The scale of the y-axis for both birds and amphibians are similar because we standardised all the explanatory variables prior to analyses, to ensure that the estimated coefficients are all on the same scale and for easier comparison of effect sizes. In response to a similar comment above, we have now clarified that the y-axis corresponds to the estimates and 95% confidence intervals for the fixed effects predicted by a single linear mixed-effects model of endemism with all predictors across grain sizes. The revised figure legend is now clearly provided in the revised manuscript (in Lines 715-722).

- If y-axis really represents predictions, then it should be reflected in the y-axis labels. But then why would you summarize it using a mean and the bars?

RESPONSE

Regarding the y-axis of the figure, we apologize for a lack of clarity. We have revised the y-axis to reflect that these are estimates and 95% confidence intervals for the fixed effects predicted from the linear mixed-effects model, which predicts endemism as a function of all predictors (temperature variation, precipitation variation, productivity variation, elevation variation, spatial autocovariate and random effects of continent identities) across different grain sizes. We have revised our description of the figure legend as follows:

“Fig. 4 | The estimates and 95% confidence intervals for the fixed effects predicted by a single linear mixed-effects model of endemism with all predictors (temperature variation, precipitation variation, productivity variation, elevation variation, spatial autocovariate and random effects of continent identities for each grid cell) across grain sizes. a birds, and b amphibians. These models indicate that our findings are explained by environmental heterogeneity at coarser grains and to a lesser extent at finer resolutions. Significance was assessed by comparing likelihoods of the fitted objects. Source data are provided as a Source Data file.”

3. Perhaps most importantly of all: You state in the caption: "These models indicate that our findings are explained by environmental heterogeneity at finer grains, and to a far lesser extent at coarser resolutions." I fail to see how to see this is actually the figures. If the y-axis are the predictions, then it makes little sense. If y-axis are effect sizes (some sort of model coefficients), then it also seems weird. What I see is that at fine grains your colourful lines (whatever they mean I have no idea) are close to 0, and they become much more variable at coarse grains. But how does this relate to your statement?

I apologize if my criticism sounds harsh, but I just can't get rid of the feeling that this part of the ms is rushed and confused. At the same time, this is actually a routine modelling exercise, and for an average statistician this would be an hour of work, including writing the methods properly. Please take the time, or ask a statistician for help, and get this really right.

RESPONSE

Thank you for these suggestions in this version. It is important to us that our methodology is clear for all readers and due to the many comments by the reviewer, we believe that the new version is much easier to follow for other readers. Because we have rewritten large parts of this section of the manuscript, we have revised our interpretation of the findings in the abstract and results to reflect that "these models indicate that our findings are explained by environmental heterogeneity at coarser grain and to a lesser extent at finer resolutions". This information is now clearly provided in the revised manuscript (in Lines 28-30, and 227-238):

[ABSTRACT]

"These findings are explained by environmental heterogeneity at coarser grains, and to a far lesser extent at finer resolutions."

[RESULTS]

"We performed these analyses across grain sizes using linear mixed-effects model with a spatial covariate to account for spatial autocorrelation (Fig. 4). Across clades, our results indicate that, in general, the explanatory power of environmental factors increases with increasing spatial grain for both weighted endemism and phylogenetic endemism and are particularly strong at the two coarsest grains of 400 and 800 km (Fig. 4). For instance, at 800 km, variation (i.e. standard deviation) in precipitation and temperature offer strong predictions of avian weighted endemism (precipitation: $\beta = 0.32$, $p < 0.001$; and temperature: $\beta = 0.19$, $p = 0.049$). The same is true for phylogenetic endemism, which shows strong relationships at intermediate to coarse grains (200 to 800 km) and is lowest for fine-grained assemblages (Fig. 4), with strong relationships of productivity and precipitation to avian phylogenetic endemism and precipitation to amphibian weighted and phylogenetic endemism."